# MLGLP: Multi-Scale Line-Graph Link Prediction Based on Graph Neural Networks

## Abstract

Scale invariance is a critical property for deep models, enabling them to consistently recognize patterns across varying resolutions or granularities. In this paper, we leverage this principle by proposing a multi-scale link prediction approach based on Graph Neural Networks (GNNs). The proposed method, Multi-Scale Line-Graph Link Prediction (MLGLP), learns graph structure and extracts informative edge representations to address information loss, capture multi-scale structural information, and improve robustness when informative node features are limited or unavailable. This approach utilizes embedding vectors generated by GNNs from enclosing subgraphs. While expanding GNN layers can capture more intricate relations, it often leads to over-smoothing, where node representations become overly similar after many layers. To mitigate this issue, we propose constructing coarse-grained graphs at three distinct scales to uncover complex relations. To apply multi-scale subgraphs in GNNs without using pooling layers that lead to information loss, we convert each subgraph into a line-graph and reformulate the task as a binary node classification problem. The hierarchical structure facilitates exploration across three levels of abstraction, fostering deeper comprehension of the relationships and dependencies inherent within the graph. We perform extensive experiments on several well-known benchmarks and compare the results with a diverse set of representative link-prediction baselines. Experiments on eight benchmark datasets demonstrate that MLGLP achieves strong AP and AUC performance and consistently benefits from combining multiple structural resolutions of local enclosing subgraphs.

## 1 Introduction

Many real-world systems can be naturally represented as graphs, in which nodes denote entities and edges describe their interactions or relationships. Graph representations are widely used in domains such as social networks, biological systems, recommender systems, chemistry, citation networks, and power grids (Cai et al., 2021). Link prediction is a fundamental graph-learning task that aims to estimate the likelihood of an unobserved connection between a pair of nodes, with applications including friend recommendation, knowledge-graph completion, and the prediction of biological interactions (Zhu et al., 2023). Traditional link-prediction heuristics, such as Common Neighbours (CN) and Adamic–Adar (AA), estimate link likelihood using predefined structural patterns. Although computationally simple and interpretable, these methods are based on fixed assumptions and may not adapt well to the structural characteristics of different networks. Graph representation-learning methods, particularly Graph Neural Networks (GNNs), instead learn predictive representations by combining graph connectivity with available node or edge information through iterative message passing.

GNN-based link prediction methods are categorized into node-based and subgraph-based approaches (Zhang et al., 2020). Node-based techniques like Graph Convolutional Networks (GCNs) (Kipf & Welling, 2017), Graph Attention Networks (GATs) (Veličković et al., 2017), GraphSAGE (SAGE) Hamilton et al. (2017), and Graph AutoEncoders (GAEs) (Kipf & Welling, 2016) incorporate multi-hop graph structures through message passing. They first extract the node embeddings and then predict the possible link between two nodes using a similarity method or a classifier, such as a multi-layer perceptron, to process both nodes'

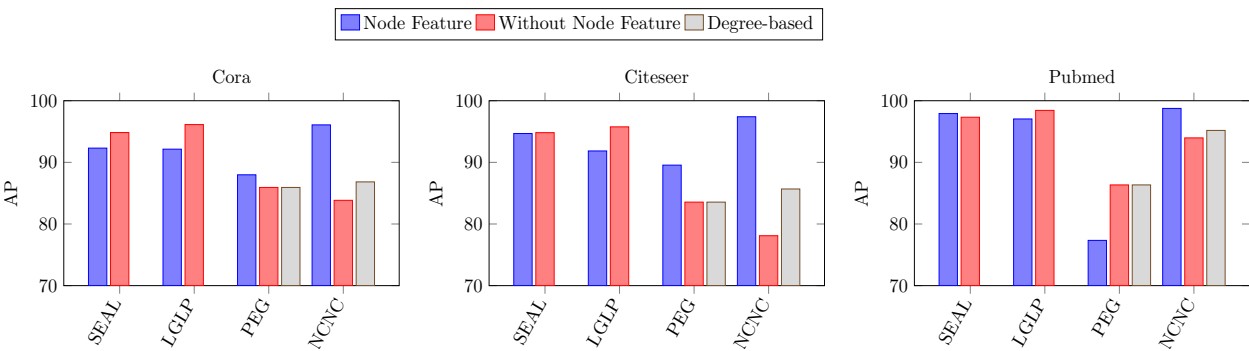

Figure 1: Effect of the presence/absence of node features on different methods (SEAL, LGLP, PEG, NCNC) for AP scores on Cora, Citeseer, and Pubmed

representations and determine the likelihood of a link between the two nodes. In contrast, subgraph-based methods, such as SEAL (Zhang & Chen, 2018), mLink (Cai & Ji, 2020), LGLP (Cai et al., 2021), PWLP (Ranjbar et al., 2026), and LGCL (Zhang et al., 2023) explicitly model the structural context of each target node pair. These methods extract an h-hop enclosing subgraph around the target link, learning a representation specifically tailored to that subgraph. They then use a binary classifier to determine whether the subgraph indicates the presence or absence of a link (Li et al., 2024). Despite their effectiveness, enclosing-subgraph methods typically represent each candidate link using a single structural resolution. The choice of the enclosing-subgraph radius introduces an inherent trade-off: a small radius may omit informative higher-order dependencies, whereas a larger radius can introduce weakly relevant or repetitive structures and increase computational cost. Increasing the number of GNN layers can expand the effective receptive field; however, deeper message passing may suffer from over-smoothing, making it difficult to preserve discriminative signals from increasingly distant nodes. Multi-scale methods such as mLink (Cai & Ji, 2020) provide hierarchical information retrieval capabilities. However, they rely on pooling layers to handle varying node numbers in subgraphs, which may result in information loss.

Recently, methods such as NCNC (Wang et al., 2023), PEG (Wang et al., 2022), and BUDDY (Chamberlain et al., 2022) have achieved strong performance in link prediction by combining node information with structural features. However, the availability and usefulness of node attributes vary considerably across graph datasets. PEG (Wang et al., 2022), for example, combines node features with positional encodings, whereas NCNC (Wang et al., 2023) and BUDDY (Chamberlain et al., 2022) exploit learned or explicitly constructed neighbourhood information. Their effectiveness under different feature settings may therefore depend on the dataset, the informativeness of the node attributes, and how missing attributes are represented. In contrast, methods such as SEAL and LGLP can operate using structural labels alone. This distinction is important for unattributed graphs, where meaningful node attributes are unavailable and a model must rely primarily on graph topology. These methods, while capable of utilizing node features, often perform better when relying solely on structural information. In fact, in these methods, incorporating both node and structural features can sometimes lead to a decline in their overall effectiveness (Cai et al., 2021). This suggests that although these methods have the capability to leverage node features, their core strength lies in capturing structural patterns, which play a more crucial role in achieving high performance.

To examine this issue empirically, we evaluate SEAL, LGLP, PEG, and NCNC on Cora, Citeseer, and Pubmed under three input settings: (1) the original node attributes, (2) randomly generated node features used as non-informative input features, and (3) node-degree features used as simple structural attributes. Figure 1 reports the resulting average precision (AP). In our experiments, PEG and NCNC exhibit larger performance reductions when the original node attributes are replaced by random features, whereas SEAL and LGLP remain comparatively robust. These results suggest that explicitly encoded pair-specific structural patterns can provide useful predictive information when informative node attributes are absent.

**Present work**. Motivated by this observation, we propose Multi-Scale Line-Graph Link Prediction (ML-GLP), a structure-centric framework that enhances link prediction by learning edge representations across

multiple structural scales. MLGLP extracts enclosing subgraphs around target node pairs and constructs coarse-grained variants of each subgraph to capture both local neighborhood patterns and broader structural dependencies. Each scale captures information at a different level of granularity. This allows the model to understand relationships between nodes at multiple levels of detail. Each scaled subgraph is then transformed into a line-graph, where edges in the original subgraph become nodes in the derived graph. This transformation reformulates link prediction as a binary node classification problem on line-graphs, enabling the model to learn edge-centric representations while preserving structural information. A GCN is applied to each scaled line-graph, and the resulting target-edge embeddings are concatenated and passed to an MLP classifier for final prediction. Hierarchical structures resulting from different graph scales, enable the analysis of graphs at different levels of granularity, allowing us to group nodes based on their relationships and capture broader patterns. Unlike SEAL and LGLP, which primarily rely on single-scale enclosing-subgraph representations, MLGLP captures collective interactions across multi-scale subgraphs and clusters, enabling richer structural information extraction and more expressive edge-centric representations for link prediction. The results obtained from experiments on real-world datasets demonstrate the superiority of the proposed method compared to state-of-the-art approaches. The main contributions of this work are summarized as follows:

1. Our method constructs multiple resolutions of each enclosing subgraph, enabling the model to capture complementary fine- and coarse-grained local structural patterns around a candidate link.

2. Our approach reformulates the link prediction problem as a node classification task using line-graphs. This approach helps in reducing information loss and simplifying the learning process for the model.

3. The proposed approach captures collective interactions within multi-scale subgraphs, enabling richer edge-centric feature extraction and more expressive structural representations for link prediction.

4. We demonstrate through extensive experiments on benchmark datasets that MLGLP consistently improves link prediction performance over heuristic, embedding-based, node-based GNN, and subgraph-based GNN baselines.

## 2 Related work

Several approaches have been proposed to address the link prediction task, which can be classified into proximity-based (non-learning) and learning-based approaches. Proximity-based methods rely on statistical properties of nodes/edges within the graph without explicitly learning embedding of nodes or edges. They typically use heuristic or feature-based techniques. Heuristic methods generally employ predefined rules or measures and evaluate link existence by assigning scores derived from the graph structure, utilizing either common neighbors or path information. Examples include Common Neighbour (CN) (Newman, 2001), Adamic Adar (AA) (Adamic & Adar, 2003), Resource Allocation (RA) (Zhou et al., 2009), Significant Influence (SI) (Yang et al., 2018), Shortest Path (SP), and Katz (Katz, 1953). They often assess the existence of a link by assigning a score derived from the graph structure. CN, AA, and RA methods primarily depend on common neighbors, whereas SI, Shortest Path, and Katz methods utilize the graph paths. These methods are widely used due to their simplicity and interoperability. However, each heuristic relies heavily on an underlying assumption regarding the likelihood of two nodes forming a connection, which constrains their efficacy when these assumptions are not met in certain network contexts. Moreover, they rely solely on graph structure and overlook node or edge features, often effective in link prediction tasks. Feature-based methods, on the other hand, use machine learning models trained on a set of features, such as node, edge, or graph attributes. While they incorporate explicit features, they may not fully exploit the underlying graph structure, potentially missing important relational information and dependencies. This can lead to less accurate or insightful models than those that utilize the graph structure directly.

Representation learning methods transform the graph structure into a low-dimensional vector space. They are divided into embedding-based and GNN-based methods. Popular embedding-based methods include Matrix Factorization (MF) (Menon & Elkan, 2011), MLP, Large-scale Information Network Embedding (LINE) (Tang et al., 2015), DeepWalk (Perozzi et al., 2014), and node2vec (Grover & Leskovec, 2016). For

instance, DeepWalk (Perozzi et al., 2014) uses the random walk strategy to generate node sequences and applies the Skip-gram model to learn node embeddings. node2vec (Grover & Leskovec, 2016) method extends the DeepWalk by using a biased random walk to better explore neighborhoods. The LINE (Tang et al., 2015) approach captures both first-order and second-order proximities in the graph for better embedding quality. A key limitation of these methods is their inability to leverage node features, relying solely on graph structure. Furthermore, they learn node embeddings with free parameters from the observed network in a transductive manner, meaning they cannot generalize to new nodes or networks not seen during training.

GNN-based methods leverage both network structure and node features. For the link prediction task, GNN-based approaches can be broadly divided into two categories: node-based and subgraph-based methods. In the Node-based category, models like GCN (Kipf & Welling, 2017), Graph Attention Networks (GAT) (Veličković et al., 2017), GraphSAGE (SAGE) (Hamilton et al., 2017), and Graph Autoencoders (GAE) (Kipf & Welling, 2016) represent nodes by leveraging the multi-hop structure of the graph through a message-passing mechanism. GCN utilizes convolutions operations on graphs to aggregate information from neighbors and learn node embedding. GAT assigns varying importance to neighbors using attention mechanisms when aggregating information. GAE uses an Encoder-Decoder framework to learn node embeddings, where the encoder maps nodes to embedding, and the decoder reconstructs the graph structure. GraphSAGE samples and aggregates features from a node's local neighborhood using neural networks.

Recent research studies have focused on subgraph-based methods, which integrate GNNs with enclosing subgraphs extracted from target node pairs, demonstrating remarkable effectiveness. The Weisfeiler-Lehman Neural Machine (WLNM) (Zhang & Chen, 2017) was among the first to apply subgraph-based GNN approaches for link prediction(Zhang & Chen, 2018). Subgraph-based methods integrate additional information, such as subgraph features and common neighbor information to gain a deeper understanding of the relationships between nodes in predicted links. Well-known methods in this category include SEAL (Zhang & Chen, 2018), BUDDY (Chamberlain et al., 2022), mLink (Cai & Ji, 2020), LGLP (Cai et al., 2021), LGCL (Zhang et al., 2023), DE-GNN (Li et al., 2020), and NBFNet (Zhu et al., 2021). The subgraph-based methods, such as SEAL (Zhang & Chen, 2018), extract an h-hop enclosing subgraph around the target link, learning a representation tailored to that subgraph.

## 3 Preliminaries

In this section, we state the problem of link prediction and provide the formal definitions for the concepts of graphs, h-hop enclosing subgraph, line-graph, Multi-scale graph.

**Graph.** Let $G = (V, E, \mathbf{X})$ be an undirected graph, where $V = \{1, \ldots, n\}$ is the node set, $E \subseteq \{\{u, v\} \mid u, v \in V, \ u \neq v\}$ is the observed edge set, and $\mathbf{X} \in \mathbb{R}^{n \times d}$ is an optional node-feature matrix. The adjacency matrix is denoted by $\mathbf{A} \in \{0, 1\}^{n \times n}$, where $A_{uv} = 1$ if $\{u, v\} \in E$, and $A_{uv} = 0$ otherwise.

**Link Prediction Problem.** The link prediction (LP) task is framed as designing a link predictor that operates on an observed subgraph $G \subset G^*$, defined as $\text{LP}(G) = \Pi : V \times V \to \{\text{True}, \text{False}\}$, which classifies the existence of links in the set of candidate edges $E_c$. The goal of LP is to estimate the likelihood of a potential connection between two nodes, $u$ and $v$, leveraging both the structural characteristics of the graph and the feature information provided by $\mathcal{A}$. Mathematically, this can be expressed as $p(u, v) = p(u, v | G, \mathbf{X})$, where $\mathbf{X}$ is the node feature matrix derived from the diagonal entries of $\mathcal{A}$. While traditional methods relied on heuristic approaches to estimate $p(u, v)$, contemporary techniques employ a learnable function $f$, parameterized by $\Theta$, enabling a more flexible and data-driven estimation: $p(u, v) = f(u, v | G, \mathbf{X}, \Theta)$. These advanced methods, often implemented through GNNs, capture complex patterns in the graph structure and feature representations, improving their ability to identify potential links, specifically true missing links. The main objective is to create a vector for each edge in the graph, which captures relevant features or characteristics of the nodes and their relationships. This vector is then fed into a binary classifier that predicts the likelihood of the presence of a given edge, or in other words, predicts whether a target node pair is likely to be connected by a true missing link in the future while avoiding misclassification of false missing links. To train the classifier, two sets of edges are used: positive and negative samples. Positive samples

are those edges that currently exist in graph G, while negative samples are a set of pairs randomly sampled from the graph where no edge currently exists.

**H-hop enclosing subgraph**. The h-hop enclosing subgraph for a node pair $(u, v)$ is the subgraph induced by the set of nodes within $h$-hops of either $u$ or $v$, i.e., nodes that are at most $h$-hops away from either $u$ or $v$. Specifically, the h-hop enclosing subgraph $G_{(u,v)}^h$ is represented as $G_{(u,v)}^h = (V_{(u,v)}^h, E_{(u,v)}^h)$, where $V_{(u,v)}^h$ is the set of nodes within $h$-hops of $u$ or $v$, and $E_{(u,v)}^h$ is the set of edges between these nodes in the original graph. The node set $V_{(u,v)}^h$ consists of all nodes $x \in V$ that satisfy $V_{(u,v)}^h = \{x \in V : d(x, u) \leq h \text{ or } d(x, v) \leq h\}$, where $d(x, y)$ represents the shortest-path distance between nodes $x$ and $y$ in the graph $G = (V, E)$. This set includes all nodes that are reachable from either $u$ or $v$ within $h$-hops. Additionally, the edge set $E_{(u,v)}^h$ includes all edges $(x, y) \in E$ such that both $x$ and $y$ belong to $V_{(u,v)}^h$, Formally expressed as $E_{(u,v)}^h = \{(x, y) \in E : x, y \in V_{(u,v)}^h\}$. These edges represent connections between nodes within the $h$-hop neighborhood of $u$ and $v$.

**Multi-scaled graph**. A multi-scaled graph $SG = (V_s, E_s)$ can be defined through several key steps, starting with the original graph $G = (V, E)$, where $V$ is the set of nodes and $E$ is the set of edges. The first step involves a coarse-graining process, in which a similarity measure $S : V \times V \to \mathbb{R}$ is defined to quantify the similarity between pairs of nodes. Various similarity measures can be utilized; specifically for the link prediction task, the similarity of a group of nodes is determined by their proximity to the target nodes. Next, a partitioning function $P : V \to C$ is established to assign each node $v \in V$ to a cluster $c \in C$ based on the similarity measure, where $C$ represents a set of clusters or hyper-nodes, denoted as $C = \{C_1, C_2, \ldots, C_k\}$. Nodes are then grouped into hyper-nodes according to predefined criteria, such that $\forall u, v \in V$, if $S(u, v) \geq \theta$, then $P(u) = P(v)$, where $\theta$ is a threshold for similarity. The specific similarity function and coarsening criterion adopted in MLGLP are introduced in Section 4. The vertex set of the scaled graph $SG$ contain the hyper-nodes, defined as $V_s = \{c_i : c_i \in C\}$. The edge set $E_s$ is constructed by connecting hyper-nodes that represent original nodes sharing edges in $G$, expressed as $E_s = \{(c_i, c_j) : \exists u \in c_i, \exists v \in c_j, (u, v) \in E\}$. Finally, the process can be recursively repeated to create multiple scales $SG_1, SG_2, SG_3$ for different levels $l$ in the hierarchy, where the $l$-th scaled graph is defined as $SG_l = (V_s^{(l)}, E_s^{(l)})$. Each level $l$ represents a different granularity of the original graph, capturing varying patterns of interactions. Transferring a graph to a new scale reduces the complexity of the graph while preserving structural information. Fig 2 shows the process of generating graphs in different scales. The

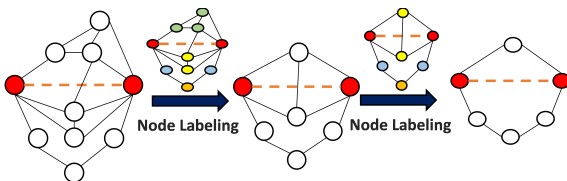

Figure 2: Example of generating graph in different scales. Similar nodes group together.

hierarchical structure of a graph enables analyzing the graph at varying granularity levels. Rather than treating every individual node independently, we can group nodes based on their relations, allowing us to capture multiscale patterns and features that might be more insightful for classification tasks. This approach enables extracting richer information by considering not just individual nodes, but also their collective interactions within subgraphs or clusters, leading to a more nuanced understanding of the graph's structure and content for classification purposes. When a node's representation closely mirrors that of its local neighborhood, it struggles to gather information effectively from distant neighbors. Consequently, the surrounding subgraph may contain repetitive or unnecessary details, which can negatively impact the performance of models designed for link prediction. In simpler terms, if a node's features are too similar to those of its nearby nodes, it may miss out on important information from farther away, potentially leading to less accurate predictions in link prediction tasks.

**Line graph.** Given an undirected graph $G = (V, E)$, its line graph is denoted by $L(G) = (V_L, E_L)$. Each node in $V_L$ corresponds to an edge in the original graph, such that $V_L = E$. Two distinct nodes in the line graph are adjacent if and only if their corresponding edges in $G$ share at least one endpoint. Accordingly, the

edge set of the line graph is defined as $E_L = \{\{e, e'\} \subseteq E \mid e \neq e' \text{ and } e \cap e' \neq \varnothing\}$. For example, if $e = \{u, w\}$ and $e' = \{w, v\}$ are two edges in $G$, then their corresponding nodes in $L(G)$ are adjacent because the two edges share the endpoint $w$. Fig. 3 illustrates the line-graph derived from the original graph.

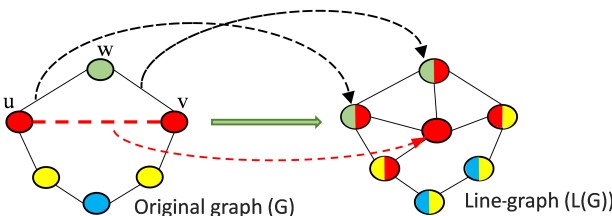

Figure 3: Demonstration of the line-graph conversion process. In the line-graph, each node corresponds to a specific edge in the original graph and is labeled with the identifiers of its two endpoints.

This approach involves representing a link as a new node in a graph and then calculating the representation of this new node to serve as a proxy for the original link. This concept, known as line-graph transformation, treats edges (links) in the original graph as nodes in a derived line-graph, thereby capturing relationships between edges through the derived graph's node representations. In this study, inspired by a method proposed in (Cai et al., 2021), we utilized the line-graph to convert the link prediction task to the node classification by using the line-graph.

## 4 Proposed methodology

In this section, we introduce the Multi-Scale line-graph Link Prediction (MLGLP) framework, depicted in Fig. 4. The first step is to group nodes with similar characteristics and connections. This consolidation allows us to transform the graph into a new scale, enabling a more efficient representation. Next, we convert the graph into a line-graph, creating three distinct line-graphs at different scales, which provide rich hierarchical structural information. Using GCN, we implement a message-passing mechanism to capture local collaborative patterns among nodes. We then focus on the embedding of target nodes within each line-graph, corresponding to target edges in each multi-scale graph derived from the original graph. By concatenating these embedding vectors, we reframe the problem from binary graph classification into a binary node classification task. To accomplish this transformation, we employ two fully connected layers as a binary classifier. We introduce a novel GNN designed to learn comprehensive relationships and features from subgraphs at different scales. This approach captures a deeper understanding of the underlying structure and dynamics of the data. The following section describes the MLGLP method in detail.

### 4.1 Subgraph extraction

Detecting the presence of a link between two nodes relies on examining the topology of the graph centered around them. While leveraging global graph structural information often improves the performance, subgraph-based methods typically limit the 2-hop neighbours to balance performance and computational cost. We extract a subgraph containing the target nodes, along with all nodes connected to them within a distance of 1 or 2. For a candidate pair $e^* = (u, v)$, we first remove $e^*$ from the observed graph and extract its h-hop enclosing subgraph. After structural labeling and multi-scale construction, the same target edge is inserted for both positive and negative candidates solely to create a designated target node in each line graph. Because this edge is inserted identically for both classes, its presence does not reveal the sample label.

### 4.2 Node aggregation - multi-scale graph transformation

Following existing GNN-based link prediction models, e.g. Zhang & Chen (2018), after extracting the h-hop enclosing subgraph $G^h_{(v_i, v_j)}$ of target pair node $(v_i, v_j)$, we map $G^h_{(v_i, v_j)}$ to three different structural scales and form coarse-grained graphs $SG_1, SG_2, SG_3$.

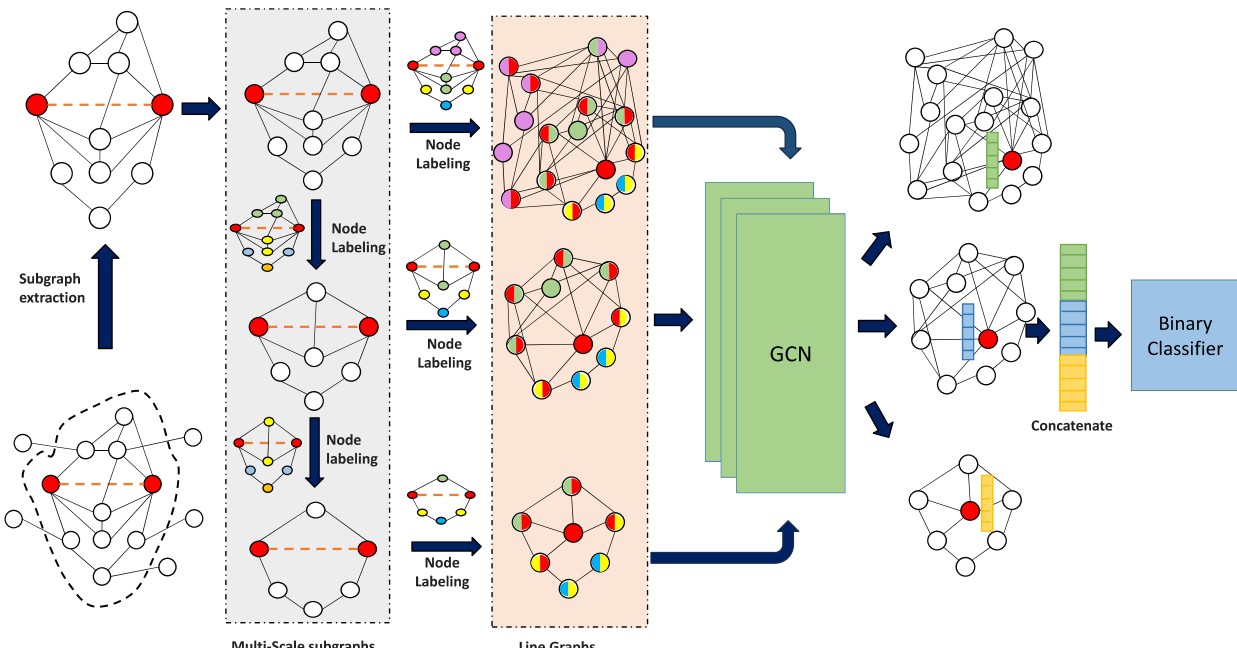

Figure 4: The overall structure of the MLGLP Framework. The process begins by extracting the enclosing subgraph from target pair nodes. We then group nodes with similar characteristics and connections, effectively merging them into single nodes. These graphs are then transformed into a line-graphs, generating three distinct line-graphs at varying scales. Using GCNs, a graph-based message-passing mechanism captures local collaborative patterns among nodes. We implement embeddings of target nodes corresponding to target edges within each multi-scale graph derived from the original graph. These embedding vectors are concatenated, reframing the problem from binary graph classification to binary node classification. A binary classifier, implemented with two fully connected layers, is employed to learn comprehensive relationships and features of subgraphs at different scales.

To ensure that the coarsening procedure remains applicable when node attributes are unavailable or uninformative, node merging is determined exclusively by structural roles relative to the target-node pair, rather than by external node features. First, each node $u$ is assigned a structural label $f(u)$ based on its shortest-path distances to the two target nodes. The two target nodes are assigned label 1. For two neighboring non-target nodes $u$ and $v$, we define $S(u, v) = \mathbb{I}\left[f(u) = f(v)\right]$, where $\mathbb{I}[\cdot]$ denotes the indicator function. We fix $\theta = 1$, and therefore merge $u$ and $v$ only when $S(u, v) \geq \theta$, which is equivalent to $f(u) = f(v)$. During merging, the incident edges of the removed node are transferred to the retained representative node, thereby preserving connections between the merged structural region and the remaining graph. The target nodes are never merged. No raw node-feature averaging, summation, concatenation, or pooling is performed.

The resulting coarsened graph is then relabeled according to its own topology. Repeating this merging-and-relabeling operation produces progressively coarser structural scales. Consequently, each scale provides a distinct structural representation of the same target-node pair.

We assign node labels using equation 1.

$$f_i(u) = \begin{cases} 1, & u \in \{v_i, v_j\}, \\ 1 + \min(d(u, v_i), d(u, v_j)) + d(u, v_i) + d(u, v_j), & \text{otherwise.} \end{cases} \tag{1}$$

where the variables $d(u, v_i)$ and $d(u, v_j)$ represent the shortest distances between a node $u$ and target-node pair, $v_i$ and $v_j$, respectively. Nodes are merged according to exact structural-label equality, after which structural labels are recomputed on the resulting coarsened graph. This process is repeated to generate the required structural scales.

### 4.3 Line-graph transformation

In this stage, we transform subgraphs $SG_1$, $SG_2$, $SG_3$ into three line-graphs $L(SG_1)$, $L(SG_2)$, $L(SG_3)$, enriching our understanding of the hierarchical structural information within the data. Each node's label in the subgraph is derived from the computation defined by equation 2, and these labels are then encoded as one-hot vectors. These labels are recomputed separately for every scaled graph. Thus, the structural representation of a node is determined by its role in the topology of that particular scale, rather than inherited by averaging the representations of the original nodes merged into it.

$$h_l(u) = 1 + \min\left(d(u, v_i), d(u, v_j)\right) \\ + \left\lfloor \frac{d(u, v_i) + d(u, v_j)}{2} \right\rfloor \left[ \left\lfloor \frac{d(u, v_i) + d(u, v_j)}{2} \right\rfloor + (d(u, v_i) + d(u, v_j)) \bmod 2 - 1 \right]. \tag{2}$$

The *labelling function* must be able to distinguish and identify two specific nodes within the subgraph. It should also be able to assign a label that reflects how important or relevant each node is in relation to the two target nodes. This involves considering the position and role of each node in the overall structure of the subgraph. Next, each subgraph is transformed into a *line-graph*, where subgraph edges become nodes in the line-graph. Then we assign a label to each node of the line-graph and use them as the initial features. This process is applied for every edge in the original graph using graph transformation function $T(.)$ in equation 3. This equation is applied after coarsening and structural relabeling to construct the initial feature of each node in the line graph. Each line-graph node corresponds to an edge $(u, w)$ in the scaled graph, and its feature is derived from the structural labels of the two endpoints.

$$T(v_i, v_j) = \text{Concat}(\text{OneHot}(\min(h_l(u), h_l(w))), \text{OneHot}(\max(h_l(u), h_l(w)))) \tag{3}$$

where $u$ and $w$ denote the two endpoints of an edge. The $\text{Concat}(\cdot)$ operation represents the concatenation of the two inputs, combining their information into a single feature vector. We merge the two one-hot vectors of nodes into a single, order-invariant vector to create their feature representation. This approach allows the edge attributes to be used as node attributes in the line-graph, thereby preserving the structural information.

### 4.4 Loss function

We apply GCN to each line-graph to generate the node representations. We focus on the embeddings of target nodes within each line-graph, particularly those corresponding to target edges for pairs of nodes within each multi-scale graph. The scale-specific representations can be fused either by direct concatenation or after learnable linear transformations. We use learnable transformations to align the representations before concatenation. Therefore, by concatenating these embeddings, the link prediction task is transformed into a binary node classification task. Node embeddings in the line-graph allows us to predict the likelihood of a potential link in the network, framing the task as a binary node classification problem. To achieve this, we employ two fully connected layers as a binary classifier.

Here we use binary cross-entropy as an objective function to treat the link prediction task as a binary classification problem. The training process minimizes the cross-entropy loss across all training links. The loss function is defined as:

$$L = -\sum_{t=1}^{N} \left( y_t \log(\hat{y}_t) + (1 - y_t) \log(1 - \hat{y}_t) \right) \tag{4}$$

where $N$ represents the total number of target links used for training, $y_t$ and $\hat{y}_t$ denote the true label value and predicted probability value of the $t^{\text{th}}$ sample, respectively, indicating whether the link exists or not. The function $\log(\cdot)$ corresponds to the natural logarithm. The pseudocode of the MLGLP algorithm and its computational complexity analysis are provided in Appendices B and G, respectively.

# 5    Performance of MLGLP on benchmark datasets

In this section, we evaluated our method (MLGLP) and compared it with 14 methods spanning various categories: heuristics, embeddings, node-based GNNs, and sub-graph-based GNNs. We report Average Precision (AP), Area Under the Curve (AUC), and training loss. Details of the evaluation metrics are provided in Appendix E. Our results show that MLGLP significantly outperforms other methods, demonstrating its effectiveness in link prediction tasks. In our study, we evaluated MLGLP against heuristic approaches including CN (Newman, 2001), AA (Adamic & Adar, 2003), RA (Zhou et al., 2009), PPR , Shortest Path (Liben-Nowell & Kleinberg, 2003), Katz (Katz, 1953), embedding techniques like MLP and MF (Menon & Elkan, 2011), node-based GNN methods including GCN (Kipf & Welling, 2017), GAE (Kipf & Welling, 2016), NCNC (Wang et al., 2023), PEG (Wang et al., 2022), and subgraph-based GNN methods such as SEAL (Zhang & Chen, 2018) and LGLP (Cai et al., 2021). Detailed descriptions of the baselines are provided in Appendix D.

**Datasets.** In this study, we evaluate the MLGLP method on a set of 8 datasets including Celegans, USAir, Power, NSC, Cora, Citeseer, Pubmed, and Router. Our experiments cover graphs of different magnitudes, encompassing variations in both node count and edge connections. Our goal is to demonstrate the broad applicability of our method across diverse datasets of varying scales, affirming its versatility and efficacy in addressing real-world challenges. The characteristics and statistics of the datasets are presented in Table 8 of Appendix C, with further information provided in the same appendix.

## 5.1    Settings

In our experiments, we set test-ratio as 0.1, which means that the datasets were randomly divided to have 90% for training, 10% for testing. All learning-based methods are trained for 50 epochs. Also, the batch number is set as 50. Each experiment was repeated ten times, and the results are reported as the mean $\pm$ standard deviation. The damping factor is set to 0.05 for Katz, and 0.85 for PPR. The learning rate for MF and MLP is set as 0.01. For a fair comparison especially with SEAL and LGLP, we set all parameters as mentioned in the original papers. We use a four-layer GCN with output dimensions [32,32,32,1] and set the MLP hidden dimension to $h = 128$. All experiments were conducted on AWS EC2 ml.p3.2xlarge instances equipped with 1 NVIDIA V100 GPU, 8 vCPUs, 61 GB of RAM.

## 5.2    Results

Table 1 reports the AP results at a test ratio of 0.1. Each result is presented as the mean $\pm$ standard deviation over ten independent runs. The heuristic methods exhibit substantial dataset-dependent variation. For example, they achieve relatively strong performance on structurally distinctive datasets such as NSC and USAir, but perform considerably worse on Power, Router, Cora, and Citeseer. The embedding-based methods, particularly MF and MLP, generally obtain lower AP values, indicating that independently learned node embeddings do not adequately represent the structural context of a candidate link. Under the featureless setting considered here, the node-based GNN methods also generally underperform the subgraph-based approaches. This suggests that explicitly modeling the target-centered enclosing subgraph provides more informative pair-specific structural representations for link prediction.

MLGLP achieves the highest mean AP across all eight datasets, although the margins over LGLP vary across datasets. Compared with LGLP, MLGLP yields improvements across all datasets, with the largest gains observed on Celegans, Router, and Cora, while the improvements on USAir, NSC, and Pubmed are relatively small. Overall, these results indicate that combining multiple local structural resolutions can provide complementary structural information for link prediction, although the magnitude of the benefit depends on the characteristics of the dataset.

Table 2 reports the AUC results obtained using a test ratio of 0.1. All values are presented as the mean $\pm$ standard deviation over ten independent runs. The results show substantial variation among the heuristic and embedding-based baselines across datasets, indicating that their effectiveness depends strongly on the underlying graph structure. Under the featureless setting considered in this study, node-based GNNs also generally perform below the subgraph-based methods. In contrast, SEAL, LGLP, and MLGLP achieve

Table 1: AP comparison of MLGLP and all baselines across eight datasets using a test ratio of 0.1.

| Methods | Celegans | Power | Router | USAir | NSC | Cora | Citeseer | Pubmed |
|---|---|---|---|---|---|---|---|---|
| | | | | Heuristics | | | | |
| CN | $80.49 \pm 0.93$ | $57.17 \pm 0.29$ | $55.26 \pm 0.41$ | $92.88 \pm 1.26$ | $96.85 \pm 0.64$ | $69.79 \pm 0.81$ | $65.27 \pm 0.61$ | $63.41 \pm 0.21$ |
| AA | $84.37 \pm 0.79$ | $57.19 \pm 0.29$ | $55.33 \pm 0.40$ | $94.52 \pm 1.12$ | $96.95 \pm 0.64$ | $70.14 \pm 0.74$ | $65.41 \pm 0.60$ | $63.46 \pm 0.20$ |
| RA | $84.69 \pm 0.80$ | $57.19 \pm 0.29$ | $55.32 \pm 0.41$ | $95.06 \pm 0.98$ | $96.97 \pm 0.64$ | $70.14 \pm 0.74$ | $65.42 \pm 0.61$ | $63.45 \pm 0.20$ |
| PPR | $80.89 \pm 1.27$ | $74.58 \pm 0.82$ | $63.76 \pm 0.37$ | $87.47 \pm 1.16$ | $97.15 \pm 0.54$ | $87.44 \pm 0.73$ | $79.01 \pm 0.70$ | $85.21 \pm 0.11$ |
| SP | $66.73 \pm 1.06$ | $74.42 \pm 0.84$ | $62.15 \pm 0.71$ | $74.44 \pm 1.17$ | $96.58 \pm 0.65$ | $84.89 \pm 0.64$ | $78.17 \pm 0.51$ | $83.34 \pm 0.18$ |
| Katz | $84.28 \pm 1.36$ | $61.59 \pm 0.51$ | $62.65 \pm 0.57$ | $93.73 \pm 0.68$ | $97.71 \pm 0.43$ | $78.79 \pm 0.86$ | $73.59 \pm 0.65$ | $78.85 \pm 0.20$ |
| | | | | Embedding | | | | |
| MF | $53.78 \pm 2.02$ | $53.22 \pm 1.74$ | $66.33 \pm 1.46$ | $59.82 \pm 4.41$ | $83.42 \pm 6.88$ | $59.44 \pm 3.55$ | $62.18 \pm 3.72$ | $52.54 \pm 3.06$ |
| MLP | $62.48 \pm 2.53$ | $50.38 \pm 0.71$ | $55.88 \pm 1.09$ | $72.38 \pm 1.92$ | $74.91 \pm 2.60$ | $55.48 \pm 1.89$ | $52.82 \pm 1.77$ | $52.70 \pm 0.68$ |
| | | | | Embedding | | | | |
| GCN | $74.90 \pm 1.62$ | $56.35 \pm 2.20$ | $84.13 \pm 0.82$ | $89.16 \pm 1.01$ | $87.62 \pm 2.16$ | $67.95 \pm 1.37$ | $60.39 \pm 2.17$ | $80.54 \pm 0.46$ |
| GAE | $74.71 \pm 1.68$ | $57.31 \pm 1.64$ | $84.83 \pm 0.98$ | $89.50 \pm 0.92$ | $92.10 \pm 1.92$ | $68.32 \pm 1.17$ | $59.47 \pm 1.43$ | $81.42 \pm 0.43$ |
| SEAL | $84.37 \pm 0.83$ | $78.33 \pm 0.67$ | $92.05 \pm 0.53$ | $95.06 \pm 0.27$ | $99.43 \pm 0.13$ | $89.64 \pm 0.67$ | $89.69 \pm 0.19$ | $95.08 \pm 0.32$ |
| LGLP | $87.82 \pm 2.69$ | $89.25 \pm 1.23$ | $95.45 \pm 0.85$ | $97.94 \pm 0.76$ | $99.57 \pm 0.29$ | $94.09 \pm 1.46$ | $93.67 \pm 1.16$ | $96.71 \pm 0.31$ |
| PEG | $84.46 \pm 0.61$ | $72.88 \pm 0.06$ | $81.18 \pm 0.04$ | $95.11 \pm 0.04$ | $97.49 \pm 0.01$ | $85.66 \pm 0.06$ | $84.36 \pm 0.03$ | $86.34 \pm 0.50$ |
| NCNC | $70.57 \pm 3.50$ | $68.67 \pm 0.98$ | $84.14 \pm 0.45$ | $85.07 \pm 2.87$ | $98.63 \pm 0.59$ | $83.84 \pm 0.65$ | $78.10 \pm 0.81$ | $93.97 \pm 0.65$ |
| MLGLP-Concat (Shared) | $89.69 \pm 3.57$ | $89.64 \pm 1.28$ | $95.54 \pm 0.38$ | $97.71 \pm 0.68$ | $99.53 \pm 0.33$ | $94.62 \pm 1.34$ | $93.78 \pm 1.15$ | $95.74 \pm 0.35$ |
| MLGLP-Concat (Independent) | $89.79 \pm 3.82$ | $89.73 \pm 1.38$ | $96.18 \pm 0.34$ | $97.91 \pm 0.68$ | $99.32 \pm 0.67$ | $94.64 \pm 1.32$ | $93.82 \pm 0.83$ | $95.82 \pm 0.23$ |
| **MLGLP-Hybrid** | $\mathbf{89.13 \pm 4.12}$ | $\mathbf{89.72 \pm 1.49}$ | $\mathbf{96.17 \pm 0.51}$ | $\mathbf{98.03 \pm 0.77}$ | $\mathbf{99.59 \pm 0.39}$ | $\mathbf{94.65 \pm 1.25}$ | $\mathbf{93.89 \pm 0.89}$ | $\mathbf{96.80 \pm 0.89}$ |

Table 2: AUC comparison of MLGLP and all baselines across eight datasets using a test ratio of 0.1.

| Methods | Celegans | Power | Router | USAir | NSC | Cora | Citeseer | Pubmed |
|---|---|---|---|---|---|---|---|---|
| | | | | Heuristics | | | | |
| CN | $83.01 \pm 0.85$ | $57.19 \pm 0.30$ | $55.29 \pm 0.40$ | $93.08 \pm 1.26$ | $96.89 \pm 0.64$ | $70.05 \pm 0.75$ | $65.36 \pm 0.63$ | $63.44 \pm 0.20$ |
| AA | $84.69 \pm 0.76$ | $57.19 \pm 0.30$ | $55.29 \pm 0.41$ | $94.00 \pm 1.17$ | $96.94 \pm 0.64$ | $70.11 \pm 0.73$ | $65.38 \pm 0.63$ | $63.44 \pm 0.20$ |
| RA | $85.08 \pm 0.78$ | $57.19 \pm 0.30$ | $55.29 \pm 0.41$ | $94.42 \pm 1.06$ | $96.95 \pm 0.64$ | $70.11 \pm 0.73$ | $65.38 \pm 0.64$ | $63.44 \pm 0.20$ |
| PPR | $82.29 \pm 0.73$ | $59.48 \pm 1.40$ | $43.71 \pm 0.84$ | $87.02 \pm 1.26$ | $97.26 \pm 0.54$ | $81.53 \pm 0.96$ | $73.98 \pm 0.86$ | $75.68 \pm 0.22$ |
| SP | $74.23 \pm 1.14$ | $60.84 \pm 2.00$ | $43.13 \pm 1.24$ | $81.07 \pm 1.04$ | $97.22 \pm 0.45$ | $80.47 \pm 1.13$ | $73.39 \pm 0.60$ | $74.75 \pm 0.27$ |
| Katz | $85.04 \pm 0.91$ | $61.60 \pm 0.50$ | $62.56 \pm 0.62$ | $92.36 \pm 0.71$ | $97.69 \pm 0.42$ | $78.86 \pm 0.83$ | $73.56 \pm 0.62$ | $78.84 \pm 0.21$ |
| | | | | Embedding | | | | |
| MF | $53.08 \pm 2.07$ | $52.65 \pm 1.37$ | $59.41 \pm 1.42$ | $58.73 \pm 4.37$ | $78.79 \pm 7.90$ | $57.32 \pm 2.70$ | $58.61 \pm 3.58$ | $51.29 \pm 1.74$ |
| MLP | $61.28 \pm 1.97$ | $50.16 \pm 0.59$ | $56.03 \pm 1.21$ | $72.10 \pm 1.94$ | $74.66 \pm 2.37$ | $54.20 \pm 1.52$ | $52.35 \pm 1.22$ | $53.17 \pm 0.57$ |
| | | | | GNN based methods | | | | |
| GCN | $74.38 \pm 1.56$ | $55.76 \pm 2.25$ | $82.54 \pm 0.75$ | $88.02 \pm 1.10$ | $89.49 \pm 1.59$ | $64.68 \pm 1.68$ | $58.71 \pm 2.60$ | $82.87 \pm 0.35$ |
| GAE | $73.91 \pm 1.55$ | $56.75 \pm 1.38$ | $83.29 \pm 1.06$ | $88.15 \pm 1.06$ | $93.61 \pm 1.18$ | $65.64 \pm 1.54$ | $60.07 \pm 1.45$ | $83.72 \pm 0.39$ |
| SEAL | $85.27 \pm 0.79$ | $75.74 \pm 0.28$ | $92.49 \pm 0.37$ | $95.02 \pm 0.07$ | $99.42 \pm 0.13$ | $87.76 \pm 0.46$ | $87.33 \pm 0.02$ | $94.59 \pm 0.29$ |
| LGLP | $88.79 \pm 2.10$ | $87.23 \pm 1.23$ | $95.31 \pm 0.81$ | $97.78 \pm 0.68$ | $99.45 \pm 0.26$ | $93.40 \pm 1.26$ | $91.98 \pm 1.38$ | $96.69 \pm 0.31$ |
| PEG | $85.13 \pm 0.53$ | $68.41 \pm 0.10$ | $79.89 \pm 0.05$ | $94.73 \pm 0.04$ | $96.37 \pm 0.01$ | $84.04 \pm 0.06$ | $82.19 \pm 0.05$ | $85.04 \pm 0.13$ |
| NCNC | $70.52 \pm 2.62$ | $62.31 \pm 0.78$ | $81.39 \pm 0.61$ | $87.52 \pm 3.56$ | $98.74 \pm 0.36$ | $78.18 \pm 0.66$ | $70.47 \pm 0.71$ | $93.68 \pm 0.61$ |
| MLGLP-Concat (Shared) | $89.26 \pm 1.83$ | $87.44 \pm 1.44$ | $95.41 \pm 0.35$ | $97.80 \pm 0.50$ | $99.52 \pm 0.35$ | $93.68 \pm 1.74$ | $92.14 \pm 1.06$ | $95.17 \pm 0.26$ |
| MLGLP-Concat (Independent) | $89.27 \pm 2.14$ | $87.60 \pm 1.49$ | $96.11 \pm 0.40$ | $98.00 \pm 0.59$ | $99.45 \pm 0.42$ | $93.67 \pm 1.55$ | $92.16 \pm 0.74$ | $95.47 \pm 0.54$ |
| **MLGLP-Hybrid** | $89.50 \pm 2.20$ | $87.53 \pm 1.66$ | $96.11 \pm 0.50$ | $98.04 \pm 0.70$ | $99.49 \pm 0.37$ | $93.72 \pm 1.56$ | $92.23 \pm 0.84$ | $96.74 \pm 0.42$ |

consistently strong performance by explicitly modeling the target-centered enclosing subgraph. MLGLP achieves the highest mean AUC on all eight datasets, with LGLP being the closest competing baseline. The clearest improvements are observed on Router and Celegans, suggesting that the coarsened structural views provide complementary information beyond the original-scale enclosing subgraph on these datasets.

To further examine the robustness of MLGLP under limited supervision, we conduct an additional experiment using 50% of the links for training and the remaining 50% for testing. Because masking removes the held-out positive edges from the observed graph and consequently changes the topology available for subgraph extraction, we additionally report the corresponding unmasked results in Appendix F as a protocol-sensitivity analysis. The masked setting remains the primary leakage-free evaluation protocol throughout the main paper, whereas the supplementary unmasked results illustrate how the methods respond to changes in the observable graph structure.

In sub-graph-based approaches, the primary focus is on extracting the enclosing subgraph around target nodes to effectively represent them based on the structure of each subgraph. In subgraph-based GNNs, each subgraph is treated independently as a training sample. Therefore, the presence of test edges in one subgraph does not influence other subgraphs. However, if test edges are masked, it hinders the accurate calculation of the structure of positive and negative samples, which can impact the learning process. To evaluate the impact of masking test edges, we conducted experiments analyzing the performance of sub-graph-based methods, specifically SEAL, LGLP, and MLGLP. The results in Table 3 indicate that masking the test data reduces performance and may introduce inaccuracies in learning patterns for both positive and negative samples. However, our proposed method, MLGLP, achieves competitive or superior performance across the evaluated datasets, while LGLP obtains better masked results on USAir and Pubmed in this setting.

Fig. 5a illustrates the training loss over 50 epochs. It is evident that MLGLP outperforms both LGLP and SEAL and achieves lower loss compared to other methods. This suggests that our proposed approach learns

Table 3: AP comparison between masked and unmasked test data using a test ratio of 0.2.

| Type | Methods | Celegans | Power | Router | USAir | NSC | Cora | Citeseer | Pubmed |
|------|---------|----------|-------|--------|-------|-----|------|----------|--------|
| Masked | SEAL | 83.12% | 77.73% | 91.10% | 95.33% | 99.61% | 89.03% | 88.79% | 94.87% |
| | LGLP | 88.25% | 84.66% | 93.43% | **96.21%** | 99.65% | 93.12% | 90.62% | **96.77%** |
| | MLGLP | **90.15%** | **87.01%** | **94.25%** | 96.20% | **99.78%** | **93.60%** | **90.96%** | 95.93% |
| Unmasked | SEAL | 86.63% | 86.71% | 97.31% | 96.44% | 99.65% | 94.80% | 94.82% | 97.32% |
| | LGLP | 89.38% | 93.71% | 99.09% | 97.23% | 99.79% | 96.20% | 95.86% | 98.28% |
| | MLGLP | **93.13%** | **94.96%** | **99.20%** | **98.28%** | **99.89%** | **96.23%** | **96.41%** | **98.29%** |

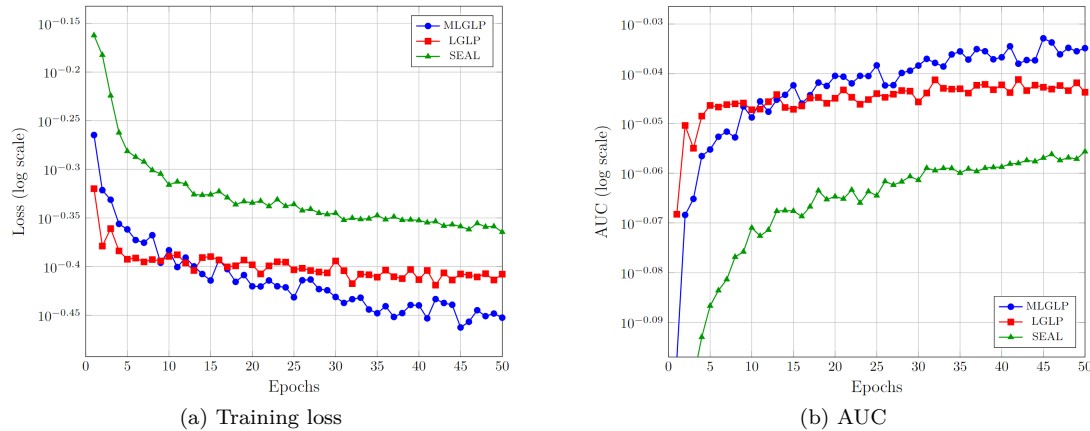

(a) Training loss            (b) AUC

Figure 5: Training loss and AUC comparison on the Celegans dataset.

more effective features for representing target links in the line-graph space. Specifically, MLGLP gathers more information from different scales during the training process, enabling it to extract complex features crucial for accurate predictions. In contrast, LGLP performs better than SEAL but does not reduce the loss as effectively as MLGLP over the training period. LGLP converges quickly but struggles to extract complex features effectively during the learning phase, as depicted in the figure. This highlights the superior capability of MLGLP in leveraging training data to enhance feature representation for link prediction tasks. Fig. 5b shows the AUC comparison between LGLP, SEAL, and MLGLP methods for Celegans dataset. The results clearly demonstrate that our proposed model significantly outperforms SEAL and LGLP in terms of achieving a higher AUC.

To highlight the performance of our proposed method, we extracted edge features from the penultimate fully connected layer and applied t-distributed stochastic neighbor embedding (t-SNE) for visualization. Fig. 6 illustrates the results on the Router, Cora, Citeseer, Power, USAir, and NSC datasets under a 0.2 test ratio, with each column corresponding to one dataset. The top row shows the visualizations from our method, while the bottom row shows those from the LGLP baseline. Positive links are depicted in Red and negative links in Blue. The visualization clearly demonstrates that the features learned by our model form well-separated clusters compared to LGLP, making the classification of positive and negative links remarkably straightforward. This showcases the strong discriminative power of our approach.

### 5.3 Ablation study

Table 4 presents the AP and AUC scores for our proposed MLGLP framework across eight datasets, evaluated using different scales and a combination of all scales in unmasked mode. The test ratio for this evaluation was set to 0.2, and we aimed to examine the contribution of each scale within the multi-scale approach. Evaluating individual scales indicates that each scale contributes valuable information, highlighting the significance of capturing different levels of structural patterns within the graph. Depending on the dataset, individual scales can sometimes yield competitive results, particularly on the NSC and Router datasets. When all scales are used together (the "All" method), the model achieves the highest performance on most

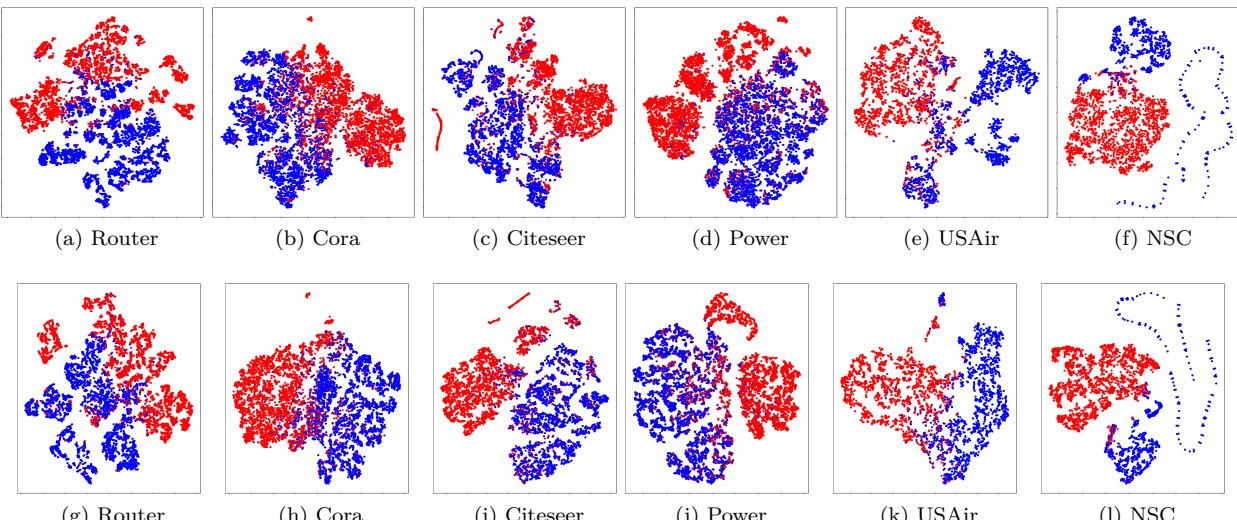

Figure 6: T-SNE visualization of edge features learned by our method (top row) and the LGLP method (bottom row) across six datasets: Router, Cora, Citeseer, Power, USAir, and NSC. Red indicates positive links; blue indicates negative links.

Table 4: Impact of different structural scales on MLGLP in terms of average precision (AP) and AUC across eight datasets using a test ratio of 0.2.

| Methods | Celegans | Power | Router | USAir | NSC | Cora | Citeseer | Pubmed |
|---|---|---|---|---|---|---|---|---|
| **AP** | | | | | | | | |
| All(MLGLP) | **93.13%** | **94.96%** | **99.20%** | **98.28%** | **99.89%** | **96.23%** | **96.41%** | **98.29%** |
| Scale-1 (LGLP) | 89.38% | 93.71% | 99.09% | 97.23% | 99.79% | 96.20% | 95.86% | 98.28% |
| Scale-2 | 88.65% | 89.14% | 97.07% | 95.87% | 99.32% | 93.43% | 93.43% | 94.36% |
| Scale-3 | 75.67% | 89.53% | 97.06% | 93.32% | 99.33% | 93.50% | 93.50% | 98.28% |
| **AUC** | | | | | | | | |
| All(MLGLP) | **90.76%** | **93.84%** | **99.11%** | **98.31%** | 99.68% | **95.79%** | **95.72%** | **98.27%** |
| Scale-1 (LGLP) | 90.75% | 92.11% | 99.05% | 98.14% | **99.82%** | 95.25% | 94.48% | 98.22% |
| Scale-2 | 88.11% | 87.93% | 97.29% | 95.75% | 99.61% | 92.35% | 92.34% | 95.24% |
| Scale-3 | 75.33% | 87.34% | 97.27% | 92.81% | 99.37% | 92.42% | 92.42% | 98.24% |

datasets. For instance, on the USAir dataset, we observe an AP of 98.28% and an AUC of 98.31%. Also the Celegans dataset achieves AP and AUC values of 93.13% and 90.76%, respectively. These results demonstrate that leveraging all scales together consistently provides the best results, confirming the robustness of the multi-scale approach.

To further justify the choice of three structural scales, we performed an ablation study by progressively combining different graph scales using the shared-encoder MLGLP architecture. The results are reported in Table 5. Scale 1 corresponds to the original line graph used in LGLP, while Scales 2–4 are constructed through successive graph coarsening. As shown in the table, using only coarse scales leads to inferior performance, indicating that each individual coarse representation loses part of the original structural information. Combining Scale 1 with Scale 2 substantially improves the results, while incorporating Scale 3 yields the best overall performance. In contrast, adding a fourth scale does not provide additional benefits and slightly degrades performance, suggesting that excessive coarsening removes useful structural information. These results support our design choice of using three structural scales.

Finally, we evaluate the effect of model capacity and parameter sharing from an efficiency perspective. Table 6 compares LGLP with two shared-encoder MLGLP configurations under the same hardware and implementation settings. The lightweight MLGLP variant with architecture $[32, 1]$ uses fewer parameters than LGLP while obtaining higher mean AP on Celegans, with a modest increase in training and inference time due to multi-scale processing. Increasing the shared encoder depth to $[32, 32, 32, 1]$ substantially increases the parameter count and runtime, but also improve predictive performance. Overally, these results suggest that

Table 5: Ablation study of different structural-scale combinations.

| Scales | Celegans | | Power | |
|---|---|---|---|---|
| | AP | AUC | AP | AUC |
| Scale 1+2+3 (MLGLP) | **89.69±3.57** | **89.26±2.15** | **89.64±1.28** | **87.44±1.44** |
| Scale 1 (LGLP) | 87.82 ± 2.69 | 88.79 ± 2.10 | 89.25 ± 1.23 | 87.23 ± 1.23 |
| Scale 2 | 84.27 ± 1.07 | 82.78 ± 0.99 | 87.38 ± 1.31 | 85.97 ± 1.38 |
| Scale 3 | 73.11 ± 1.26 | 73.41 ± 1.17 | 87.39 ± 1.42 | 85.99 ± 1.48 |
| Scale 4 | 74.72 ± 1.12 | 74.14 ± 1.34 | 87.36 ± 1.33 | 85.88 ± 1.46 |
| Scale 1+2 | 88.22 ± 2.18 | 89.19 ± 1.31 | 88.62 ± 1.21 | 86.52 ± 1.16 |
| Scale 1+3 | 87.96 ± 1.34 | 86.32 ± 1.14 | 88.06 ± 1.13 | 86.46 ± 1.11 |
| Scale 1+2+3+4 | 87.93 ± 2.26 | 86.25 ± 1.63 | 87.73 ± 1.09 | 86.57 ± 1.15 |

Table 6: Empirical efficiency comparison on the Celegans dataset. Runtime was measured under the same hardware and implementation settings.

| Method | Architecture | #Params | AP (↑) | Train / epoch (s) | Inference (s) |
|---|---|---|---|---|---|
| LGLP | [32,32,32,1] | 16,003 | 87.82 ± 2.69 | 2.94 | 0.153 |
| MLGLP Shared | [32,1] | **14,147** | 88.99 ± 2.80 | 3.86 | 0.174 |
| MLGLP Shared | [32,32,32,1] | 40,835 | **89.69 ± 3.57** | 5.86 | 0.216 |

the performance of MLGLP is not solely attributable to increased model capacity, and that the lightweight shared encoder offers a practical accuracy–efficiency trade-off.

## 6 Conclusion and future research

In this study, we propose a novel approach using GNNs called Multi-Scale Line-Graph Link Prediction (MLGLP). This method aims to effectively learn the graph structure and extract representative features from edges, addressing challenges such as information loss and handling multi-scale information. This method constructs three progressively coarsened views of each 2-hop enclosing subgraph and transforms them into line graphs. This transformation allows us to learn node embeddings within each subgraph and translates the link prediction task into a node classification problem. By combining the target-edge representations learned from these views, MLGLP captures complementary fine- and coarse-grained structural patterns within the local neighborhood of a candidate link. Experiments on eight benchmark datasets demonstrate that MLGLP achieves strong AP and AUC performance compared with a broad range of link-prediction baselines. Moreover, a shared-encoder variant maintains comparable predictive performance while reducing the number of trainable GCN encoder parameters by approximately threefold. Future work will extend MLGLP to heterogeneous graphs, where node and edge types can provide additional structural and semantic information.

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

# A   Notations

This section provides an overview of the symbols and notations utilized in this paper, with a detailed summary provided in Table 7.

Table 7: Summary of notations used in the paper

| Notations | Definitions or Descriptions |
|---|---|
| $G = (V, E, X)$ | Graph with node set $V$, edge set $E$, and node features $X$ |
| $n$ | Number of nodes, $n = |V|$ |
| $m$ | Number of edges, $m = |E|$ |
| $E^*$ | Complete set of possible edges between nodes in $V$ |
| $\mathbf{A}$ | Adjacency matrix, $\mathbf{A} \in \{0, 1\}^{n \times n}$ |
| $\mathbf{A}_{i,j}$ | Adjacency matrix entry, 1 if edge $(i, j)$ exists, otherwise 0 |
| $\mathbf{X}$ | Node feature matrix, $\mathbf{X} \in \mathbb{R}^{n \times k}$ |
| $\mathbf{X}_i$ | Feature vector for node $i$, $\mathbf{X}_i = \mathcal{A}_{i,i,:}$ |
| $G^*$ | Complete graph with all possible edges $E^*$ between nodes in $V$ |
| $E_c$ | Set of candidate edges for link prediction |
| $p(u, v)$ | Probability of a link between nodes $u$ and $v$ |
| $f(u, v|G, \mathbf{X}, \Theta)$ | Learnable function to estimate $p(u, v)$, parameterized by $\Theta$ |
| $\Theta$ | Parameters of the learnable function $f$ |
| $h$ | Maximum number of hops in the h-hop enclosing subgraph |
| $G^h_{(u,v)}$ | h-hop enclosing subgraph for node pair $(u, v)$ |
| $V^h_{(u,v)}$ | Node set within $h$-hops of either $u$ or $v$ |
| $E^h_{(u,v)}$ | Edge set within $h$-hops of either $u$ or $v$ |
| $d(x, y)$ | Shortest-path distance between nodes $x$ and $y$ |
| $SG = (V_s, E_s)$ | Multi-scaled graph with vertex set $V_s$ and edge set $E_s$ |
| $S(u, v)$ | Binary structural similarity (1 if structural labels are identical; 0 otherwise) |
| $P(v)$ | Partitioning function assigning node $v$ to a cluster |
| $C$ | Set of clusters or hyper-nodes |
| $\theta$ | Fixed similarity threshold $\theta=1$, enforcing exact structural-label equality |
| $SG_l = (V_s^{(l)}, E_s^{(l)})$ | Scaled graph at level $l$ in a hierarchical structure |
| $L(G) = (V_L, E_L)$ | Line-graph of graph $G$ |
| $V_L$ | Node set of the line-graph |
| $E_L$ | Edge set of the line-graph |
| $L$ | Loss function |
| $N$ | Total number of target links used for training |
| $y_t$ | True label value for the $t$th sample |
| $\hat{y}_t$ | Predicted probability value for the $t$th sample |
| $f_i(u)$ | Label of node $u$ |
| $h(u)$ | Node representation of node $u$ |
| $T(u, w)$ | Function for concatenating features of nodes $u$ and $w$ |

**True Missing Links** are edges that should exist in the graph but are not currently observed. These edges are part of the complete graph's true edge set $E^*$ but are not included in the observed edge set $E$. Formally, a true missing link $(u, v)$ satisfies:

$$\{(u, v) \in E^* \text{ and } (u, v) \notin E\}$$

where $E$ is the set of obsserved edges and $E^*$ is the set of all true edges in the complete graph. For example if $E^*$ includes an edge $(u, v)$ but $(u, v)$ is not in $E$, then $(u, v)$ is a true missing link.

**False Missing Links** are pairs of nodes that are incorrectly considered as potential links but are not part of the true edge set $E^*$. These are often pairs where a link is not present in both the complete graph and

the observed graph. Formally, a false missing link $(u, v)$ satisfies:

$$\{(u, v) \mid u, v \in V \text{ and } (u, v) \notin E^* \text{ and } (u, v) \notin E\}$$

for example if $E^*$ does not include an edge $(u, v)$ and $(u, v)$ is also not in $E$, this edge can be mistakenly considered a potential link (false positive) by some models.

**Positive Samples** are defined as edges that exist in both the observed edge set $E$ and the complete graph $E^*$. In other words, these are edges that are part of the true edge set and are observed. Formally, these are edges in $E \cap E^*$.

**Negative Samples** are defined as edges that do not exist in either the set of true edges $E^*$ or the set of observed edges $E$. They can be considered as false positives, i.e., pairs of nodes that should not be connected. These correspond to pairs of nodes that are not connected by an edge in the graph. Since identifying true negative samples, edges that should never exist, is often challenging due to the lack of complete knowledge about the graph or underlying data distribution, a common strategy is randomly selecting negative samples from the set of nonexistent edges. Thus, the negative samples are selected randomly from edges that are not in $E$, acknowledging that while these may not be guaranteed to be true negatives, they serve as a practical and sufficient set of negative examples for training the model.

**Candidate Set** ($E_c$) is constructed to include both types of samples to train and evaluate the link prediction model effectively. The goal is to differentiate between these two types of samples and correctly classify which candidate edges should be present in the graph. These candidates help train and test the model to accurately predict the presence of true missing links while avoiding false positives.

## B    Algorithm

In this section, we present the pseudo-code of the MLGLP framework to enhance clarity and understanding.

---
**Algorithm 1** MLGLP Algorithm
---
**Require:** Graph $G$, candidate node pair $(v_i, v_j)$, and $h = 2$
**Ensure:** Predicted probability of the candidate link
  1: Remove the candidate edge $(v_i, v_j)$ from the observed graph
  2: Extract the $h$-hop enclosing subgraph $G^h_{(v_i, v_j)}$
  3: Assign target-relative structural labels using Eq. equation 1
  4: Set $SG_1 \leftarrow G^h_{(v_i, v_j)}$
  5: **for** $s = 2, 3$ **do**
  6:     Merge neighboring non-target nodes having identical structural labels
  7:     Construct the coarsened graph $SG_s$
  8:     Recompute structural labels on $SG_s$ using Eq. equation 2
  9: **end for**
 10: **for** $s = 1, 2, 3$ **do**
 11:     Construct edge features using Eq. equation 3
 12:     Convert $SG_s$ into its corresponding line graph $LSG_s$
 13:     Apply the GCN encoder to $LSG_s$
 14:     Extract the embedding of the node corresponding to the target edge
 15: **end for**
 16: Concatenate the three target-edge embeddings
 17: Predict the link probability using the binary classifier
---

## C    Datasets

A brief overview of the benchmark datasets utilized in this study is as follows. These datasets are used for evaluating models in various types of graphs, including social networks, citation networks, and biological networks.

- Cora: This citation graph includes 2708 scientific publications and 5278 links, with a dictionary of 1433 unique words derived from the papers.

- Citeseer: This dataset features 3327 scientific publications and 4552 links, accompanied by a dictionary of 3703 unique words from the publication texts.

- Pubmed: This citation graph includes 19,717 scientific publications and 44,324 citation links, with a dictionary of 500 unique words derived from the papers.

- Router: This dataset represents a router-level Internet graph with 5022 nodes and 6258 edges, modeling connections between routers in the network.

- USAir: This dataset represents a graph of US airlines, containing 332 nodes and 2126 edges.

- NSC: This dataset illustrates the collaboration relationships of network science researchers, containing 1461 nodes and 2126 edges.

- Celegans: This dataset contains the biological neural network of *C. elegans*, consisting of 297 nodes and 2148 edges.

- Power: This dataset illustrates the topology of the Western States Power Grid of the United States, containing 4941 nodes and 6594 edges.

Table 8: Summary statistics of the datasets used in the study

| Statistic | Router | Cora | Citeseer | Pubmed | USAir | NSC | Celegans | Power |
|-----------|--------|------|----------|--------|-------|------|----------|-------|
| #Nodes | 5022 | 2708 | 3327 | 19,717 | 332 | 1461 | 297 | 4941 |
| #Edges | 6258 | 5278 | 4552 | 44,324 | 2126 | 2126 | 2148 | 6594 |
| #Features | NA | 1433 | 3703 | 500 | NA | NA | NA | NA |

## D  Baseline methods

This section provides detailed descriptions of the baseline methods utilized in this paper. We compare our proposed MLGLP with 14 methods spanning various categories: heuristics, embeddings, node-based GNNs, and sub-graph-based GNNs. In our study, we evaluated our proposed method against heuristic approaches such as CN, AA, PPR, Shortest Path, and, Katz, embedding techniques like MLP and MF, node-based GNN methods including GAE, GCN, NCNC, PEG, and finally subgraph-based GNN methods such as SEAL and LGLP. Table 9 shows Details of heuristic-based methods utilized in this paper.

Table 9: Details of heuristic-based methods for link prediction utilized in this paper

| Name | Formula | Order |
|------|---------|-------|
| Common Neighbors (CN) | $|\Gamma(x) \cap \Gamma(y)|$ | first |
| Adamic-Adar(AA) | $\sum_{z \in \Gamma(x) \cap \Gamma(y)} \frac{1}{\log |\Gamma(z)|}$ | second |
| Resource Allocation (RA) | $\sum_{z \in \Gamma(x) \cap \Gamma(y)} \frac{1}{|\Gamma(z)|}$ | second |
| PageRank(PPR) | $[\pi_x]_y + [\pi_y]_x$ | high |
| Shortest Path | $\frac{1}{length(shortestpath(x,y))}$ | high |
| Katz | $\sum_{l=1}^{\infty} \beta^l |\text{walks}^{\langle l \rangle}(x,y)|$ | high |

## E    Evaluation metrics

This section provides detailed descriptions of evaluation metrics including AUC, AP.

**a) AUC:** The AUC is computed as the number of successful predictions divided by the total number of comparisons. Successful predictions can be determined based on scores for each node pair using predefined heuristics (e.g., common neighbors) or probabilities for each node pair using the GNN model. Compute the AUC using the formula:

$$AUC = \frac{n' + 0.5 \times n''}{n} \tag{5}$$

where $n$ is the total number of predictions, $n'$ is the number of successful predictions (i.e., the number of times for each positive-negative pair the positive sample has a higher score or probability than the negative sample), and $n''$ is the number of times that scores are the same.

**b) AP:** Average Precision measures the precision of a model at various threshold levels, capturing the ability to rank positive links higher than negative ones. It is formulated as:

$$AP = \sum_{k=1}^{n} P(k) \cdot \Delta r(k) \tag{6}$$

Where $n$ is the total number of positive and negative links, $P(k)$ is the precision at rank $k$, and $\Delta r(k)$ is the change in recall at rank $k$.

## F    Robustness evaluation with reduced training data

To demonstrate the robustness of our proposed approach with reduced training data, we conducted experiments on all datasets using only 50 percent of the training links. The remaining links were used as test data. The outcomes, including AUC and AP values, are presented in Tables 10 and 11 The results consistently show that our method outperforms all baseline methods in terms of AP across all datasets and AUC in the majority of datasets. Remarkably, our method maintains strong performance even with only 50 percent of the training links, achieving AUC and AP values comparable to those obtained with 80 percent of the training links.

## G    Time complexity analysis

For extracting each graph $G^h_{(v_i,v_j)}$ with $n$ nodes and $m$ edges where $(v_i, v_j)$ is the target pair of nodes, total time complexity for calculating distances for both target nodes is $\mathcal{O}(n + m)$ per scale. Constructing a *line-graph* has a time complexity of $\mathcal{O}(m^2)$, where $m$ is the number of edges in the original graph (since each edge in the original graph becomes a node in the *line-graph*). Time Complexity for GCN operation depends on the number of nodes and edges in the line-graph. For a graph with $|V|$ nodes and $|E|$ edges, the complexity is $\mathcal{O}(|V| + |E|)$. Since we are dealing with line-graphs, this translates to $\mathcal{O}(n^2 + n^2) = \mathcal{O}(n^2)$ for each scale. For three scales, it's $\mathcal{O}(3n^2) = \mathcal{O}(n^2)$.

Table 10: AUC comparison of MLGLP and all baseline methods across eight datasets using a test ratio of 0.5.

| Methods | Celegans | Power | Router | USAir | NSC | Cora | Citeseer | Pubmed |
|---|---|---|---|---|---|---|---|---|
| **Heuristics** | | | | | | | | |
| CN | 70.87% | 53.14% | 53.39% | 87.82% | 90.38% | 59.69% | 57.71% | 57.41% |
| AA | 72.52% | 53.05% | 52.57% | 88.11% | 92.41% | 59.79% | 57.63% | 57.39% |
| RA | 72.78% | 53.28% | 52.86% | 87.77% | 92.60% | 59.12% | 58.41% | 57.28% |
| PPR | 79.51% | 57.73% | 54.32% | 85.42% | 95.56% | 68.39% | 67.29 | 73.06% |
| Shortest Path | 72.11% | 57.45% | 55.46% | 82.66% | 94.86% | 67.10% | 66.53% | 72.80% |
| Katz | 80.09% | 58.28% | 60.11% | 89.71% | 96.17% | 68.95% | 68.01% | 69.90% |
| **Embedding** | | | | | | | | |
| MF | 71.99% | 54.70% | 77.47% | 90.87% | 98.64% | 63.96% | 62.28% | 87.18% |
| MLP | 62.81% | 51.19% | 60.32% | 80.32% | 89.31% | 55.88% | 53.19% | 54.57% |
| **GNN based methods** | | | | | | | | |
| GCN | 73.80% | 52.44% | 59.49% | 90.08% | 98.09% | 60.14% | 60.40% | 65.72% |
| GAE | 62.20% | 53.88% | 49.65% | 70.70% | 76.57% | 83.44% | 62.85% | 83.75% |
| SEAL | 88.19% | 82.21% | 97.12% | 96.32% | 99.64% | 93.42% | 92.73% | 96.86% |
| LGLP | 90.94% | 91.78% | 98.98% | 97.34% | 99.77% | 95.22% | 94.35% | 98.38% |
| PEG | 73.42% | 57.67% | 61.51% | 88.16% | 84.39% | 67.48% | 65.25% | 70.41% |
| NCNC | 81.16% | 61.20% | 80.18% | 81.42% | 87.70% | 67.67% | 60.73% | 87.71% |
| MLGLP | **91.52%** | **93.28%** | **98.62%** | **97.22%** | **99.83%** | **95.33%** | **94.99%** | **98.40%** |

Table 11: AP comparison of MLGLP and all baseline methods across eight datasets using a test ratio of 0.5.

| Methods | Celegans | Power | Router | USAir | NSC | Cora | Citeseer | Pubmed |
|---|---|---|---|---|---|---|---|---|
| **Heuristics** | | | | | | | | |
| CN | 68.23% | 53.12% | 53.38% | 87.47% | 90.34% | 59.57% | 57.68% | 57.38% |
| AA | 70.72% | 53.05% | 52.55% | 88.46% | 92.44% | 59.91% | 57.64% | 57.39% |
| RA | 72.24% | 53.27% | 52.80% | 88.51% | 92.61% | 59.18% | 58.41% | 57.28% |
| PPR | 78.25% | 57.68% | 60.89% | 84.90% | 95.58% | 75.41% | 69.07% | 80.17% |
| Shortest Path | 66.00% | 57.65% | 60.94% | 77.60% | 94.47% | 73.46% | 68.70% | 79.10% |
| Katz | 79.90% | 58.42% | 61.15% | 92.03% | 96.25% | 75.71% | 69.49% | 69.88% |
| **Embedding** | | | | | | | | |
| MF | 82.25% | 54.48% | 81.38% | 91.19% | 98.86% | 66.42% | 65.66% | 87.73% |
| MLP | 64.28% | 51.19% | 60.06% | 81.35% | 89.67% | 56.84% | 54.26% | 54.10% |
| **GNN based methods** | | | | | | | | |
| GCN | 73.06% | 53.08% | 62.04% | 90.56% | 98.33% | 62.96% | 64.25% | 68.18% |
| GAE | 61.54% | 53.96% | 49.80% | 68.44% | 66.95% | 82.48% | 65.96% | 82.64% |
| SEAL | 86.58% | 85.57% | 97.06% | 95.96% | 99.51% | 94.73% | 94.43% | 97.24% |
| LGLP | 89.63% | 93.87% | 98.86% | 98.28% | 99.77% | 95.98% | 95.69% | 98.40% |
| PEG | 74.49% | 57.41% | 58.13% | 90.15% | 89.31% | 67.98% | 66.52% | 72.28% |
| NCNC | 79.93% | 68.35% | 83.47% | 74.51% | 90.13% | 74.16% | 66.14% | 88.29% |
| MLGLP | **93.43%** | **95.04%** | **99.06%** | **98.54%** | **99.92%** | **96.42%** | **95.79%** | **98.41%** |

