# OpenReview forum: "MLGLP: Multi-Scale Line-Graph Link Prediction Based on Graph Neural Networks"
_TMLR — Under review for TMLR_

### Review · Reviewer_ydQz · 2026-06-27

**Summary Of Contributions:**

The paper proposes Multi-Scale Line-Graph Link Prediction. Given a target node pair, the method extracts a 2-hop enclosing subgraph, builds three coarse-grained versions of it by merging nodes that share a distance-based label, converts each scale to its line-graph (so edges become nodes), applies a GCN to each line-graph, and concatenates the three target-edge embeddings into a two-layer MLP for binary classification. The method extends LGLP from a single scale to three scales. Experiments on eight small benchmark graphs report Average Precision and AUC against heuristic, embedding, node-based GNN, and subgraph-based GNN baselines, plus an ablation over scales, a masked-versus-unmasked study, a reduced-training-data study, t-SNE visualizations, and a complexity analysis.

### Strengths
The approach is clearly motivated and easy to follow. The supplementary code implements the described pipeline and produces sensible outputs.

### Weaknesses (more details in the next sections)
The headline results are produced with test edges left in the graph, and the central superiority claim is contradicted by the paper's own masked table. The reported metrics are saturated and the discriminative metrics the code computes are not reported. The strongest baselines are run in an unspecified and unfavorable featureless setting. Several tables are internally inconsistent, and the supplied data cannot reproduce the reported results (the most-reported dataset is absent).

**Audience:**

Yes

**Audience Explanation:**

Yes. Link prediction via line-graph transformation and multi-scale or coarsened subgraph representations is an active topic, and some members of the GNN and link-prediction community would be interested in a structure-centric method that operates without node features, and in the masked-versus-unmasked sensitivity it (inadvertently) documents.

**Broader Impact Concerns:**

None specific to the method. Link prediction on small benchmark graphs as studied here does not raise ethical concerns that would require a Broader Impact Statement. The reproducibility and data discrepancies noted above are research-quality matters addressed under Requested Changes, not broader-impact issues.

**Claims And Evidence:**

No

**Claims Explanation:**

The paper's central claims (that MLGLP "significantly outperforms" baselines, "consistently" beats SEAL and LGLP, and is robust under reduced training data) are not supported by the evidence as presented. The issues below are mostly checkable against the paper's own tables or the supplied code and data.

1. The main results leak test edges, and the key claim fails under the paper's own corrected protocol. In the released code, masking of test edges is off by default, so the full adjacency including test positives is used to build subgraphs during both training and testing. Tables 1 and 2 (the headline AP/AUC) are these unmasked numbers. The unmasked block of Table 3 reproduces them exactly. Table 3 itself shows the inflation: masking lowers MLGLP AP by up to 7.95 (Power), 5.45 (Citeseer), and 4.95 (Router). More critical, the text states MLGLP "consistently demonstrates superior performance across all datasets compared to SEAL and LGLP," but in the masked (correct) block of Table 3, LGLP beats MLGLP on USAir (96.21 vs 96.20) and Pubmed (96.77 vs 95.93). The headline claim holds only in the leaky configuration. Consistent with this, running the code in its default (unmasked) setting reaches the high 0.9s within one to two epochs (USAir test AUC/AP about 0.95 to 0.96 by epoch 1, versus a 50-epoch reported 98.31/98.28), which is what one expects when leakage plus easy negatives make the task trivial. This is the SpotTarget pitfall, which the paper cites, but never actually engaged with.

2. The metrics are saturated and the discriminative metrics computed and withheld. Evaluation uses AP and AUC against randomly sampled negatives, which saturates near the ceiling and is why nearly all methods land at 94 to 99 percent. This is the pitfall the HeaRT benchmark (also cited in the paper, but not actually taken into account) was written to address. The released evaluation code computes Hits@K and MRR and prints them every epoch during training, yet none are reported. The non-saturated metrics that would actually separate methods exist in the pipeline and are omitted.

3. The featureless comparison is both unfavorable to the baselines and unspecified. All methods are run on "plain graphs without node features," and MLGLP uses only structural distance labels (confirmed in code: node features in cora.mat and citeseer.mat are ignored). The feature-driven baselines NCNC and PEG are then reported at implausibly low scores (NCNC 71.39 AP on Celegans, 82.54 on USAir), i.e., evaluated in the regime where they are weakest. The paper never states how the node-based baselines obtain inputs when no features exist (random, one-hot, degree?). The claim of superiority over these methods is therefore not supported, and the setup is not reproducible.

4. The ablation does not establish that the claimed mechanism causes the gains. Scale-1 in the ablation equals the LGLP baseline numerically, so the contribution reduces to two added coarse scales. The improvements over LGLP are frequently within likely noise (AP: Cora +0.03, Pubmed +0.01, Router +0.11, NSC +0.10), and although experiments are described as 10 runs averaged, no standard deviations are reported anywhere. The ablation varies only the number of scales, it never removes the line-graph transformation or tests an alternative to concatenation, so the factors driving any improvement cannot be attributed. As a matter of evidence (independent of novelty, which I am not weighing per TMLR criteria), the paper does not show that multi-scale coarsening, rather than run-to-run variance, is responsible for the reported gains.

5. Internal inconsistencies in the results.

     * Appendix F states 50 percent training reduces AP by "3.45%, 1.69%, 0.58%, 2.67%, 0.06%, 0.90%" for Celegans, Power, Router, USAir, NSC, Cora. The actual Table 9 minus Table 1 deltas are +0.30, +0.08, -0.14, +0.26, +0.03, +0.19. Five of six increase, none match the stated reductions.

     * In Table 2 (AUC), three Router heuristics report AUC below 0.5 while their AP in Table 1 is above 0.6, for the same scores on balanced negatives: PPR (AUC 45.83 / AP 64.67), Shortest Path (40.74 / 61.42), Katz (43.89 / 63.85). These pairs are mutually inconsistent for one score vector. Can the authors explain how these entries were computed?

     * In Table 4, Cora and Citeseer report identical values to two decimals at Scale-2 AP (both 93.43), Scale-3 AP (both 93.50), and Scale-3 AUC (both 92.42), despite being different graphs of different sizes. Can the authors explain how independent runs on two different datasets produced identical numbers, or confirm whether these cells were duplicated?

     * Katz on Pubmed reports exactly 50.00 for both AUC (Table 8) and AP (Table 9) at the 0.5 ratio. A value of exactly 0.5 is the signature of a constant or degenerate score vector rather than a computed heuristic result. Can the authors explain how this number was produced, and whether it reflects a failed run reported as a result?

6. The supplied data cannot reproduce the reported results. Reading the .mat files directly: Pubmed is absent from the package, yet Pubmed appears in Tables 1, 2, 8, 9 and Figure 1 (a Pubmed run crashes at load). NSC has 2126 edges versus the claimed 2742, and the text and Table 6 disagree on its node count (1589 vs 1461). citeseer.mat has 3327 nodes versus the stated 3312, Cora has 5278 edges (text) versus 5429 (Table 6). An undocumented dataset (ADV, 5155 nodes) is shipped but never used, and the results directory is empty. The most-reported dataset cannot be reproduced at all.

7. The "state-of-the-art" comparison omits the relevant strong methods. The claim of outperforming state of the art is not substantiated, because methods named in the paper's own related work (BUDDY, DE-GNN, NBFNet, and LGCL, a directly comparable line-graph LP method) are never benchmarked. Among learned structural methods only SEAL and LGLP are meaningfully contested.

8. The references and writeup show signs of not being verified against sources. Equation 2 (line-graph labeling) is a malformed DRNL: it uses d(u,vi) where d(u,vi)+d(u,vj) is required (the code implements the correct version). In the text, node2vec is attributed to "Qiu et al., 2018" (which is NetMF) and GCN to "Yao et al., 2019" (which is TextGCN). The canonical node2vec and GCN papers are absent. The reference list contains multiple incorrect author names relative to the actual papers (for example the benchmarking paper, the Significant Influence paper, the line-graph contrastive paper, and SpotTarget). Individually minor, but together they indicate the manuscript and bibliography were not checked against the underlying sources, which lowers confidence in the paper.

Taken together, the evidence does not support the claims. Establishing them would require re-running the entire evaluation under a corrected protocol (masked graph, hard negatives, ranking metrics, variance), a fair baseline setting, and a reproducible data package. That is an overhaul of the empirical section rather than a revision of claims, which is why I answer No here.

**Requested Changes:**

I mark each change as [Critical] (needed before the claims could be considered supported) or [Strengthen] (would improve the work).

[Critical] Report all main results with test edges masked, i.e. removed from the message-passing graph during training and evaluation. Make masking the default and present the masked numbers as the primary results. Reconcile or retract the "consistently superior to SEAL and LGLP" claim with the masked outcomes (LGLP currently beats MLGLP on USAir and Pubmed when masked).

[Critical] Replace or supplement random-negative AP/AUC with ranking metrics under hard negatives (Hits@K and MRR, which the code already computes), reported for all methods. State the negative-sampling protocol explicitly.

[Critical] Report standard deviations over the 10 runs for every reported number, so the gaps over LGLP (often under 0.3 AP points) can be assessed against variance.

[Critical] Specify exactly how each baseline is run in the featureless setting (input features used for GCN, GAE, PEG, NCNC), and additionally evaluate the feature-based baselines (PEG, NCNC) with their intended node features. Do not characterize results as outperforming these methods without this.

[Critical] Fix the code package so the reported experiments are reproducible: include Pubmed (or remove all Pubmed results), document or remove ADV.mat, and reconcile the node/edge counts in the text and Table 6 with the actual files (NSC, Citeseer, Cora). Provide a script that regenerates the paper's tables.

[Critical] Resolve the internal inconsistencies: the reduced-data percentages in Appendix F that contradict Tables 1 and 9, the Router heuristic rows where AUC < 0.5 but AP > 0.6, the identical Cora/Citeseer cells in Table 4, and the exactly-50.00 Katz/Pubmed entries in Tables 8 and 9 (explain how each was computed).

[Critical] Extend the ablation to isolate the contributions: line-graph versus no line-graph, and the multi-scale concatenation versus single-scale, so the source of any improvement is identified rather than asserted.

[Critical] Compare against the directly relevant baselines named in the paper's own related work, at minimum LGCL, BUDDY, and NBFNet, under the corrected protocol. Critical for any "state-of-the-art" claim. Otherwise soften that claim.

[Critical] Correct Equation 2 to the intended DRNL form, fix the in-text citations for node2vec and GCN and correct the author names and categorizations in the reference list and Appendix D.

[Strengthen] Provide a working, pinned environment (a requirements.txt or environment.yml that installs). The current README pins Python 3.8.1 (end of life since October 2024) and PyTorch 1.4.0 while instructing unversioned installation of the PyG ecosystem. These are mutually incompatible and do not install as written.

[Strengthen] Honor the existing --no-cuda flag (CUDA placement is currently hard-coded, so the code cannot run CPU-only without a source edit), and align the result-directory and method-name strings (MLGLP vs MSLGLP).

[Strengthen] Reconcile the architecture description (Section 5.1 states three 32-dim conv layers, 3x32 = 96) with the implementation (four GCN layers [32,32,32,1] with concatenated layer outputs, 291 into the MLP), and add empirical training and inference times alongside the asymptotic analysis, including a comparison to SEAL and LGLP.

---

> ### Author Response · Authors · 2026-07-20
>
> We thank the reviewer for the thoughtful feedback. Below we address the concerns point-by-point.
>
> **Q1:**  We have revised the experimental protocol so that test edges are removed from the message-passing graph during training, and masking is now the default setting. Accordingly, **Tables 1 and 2** have been replaced with the masked AP and AUC results. We have also revised the corresponding discussion.
>
> **Q2:** We thank the reviewer for this important suggestion. Hits@K and MRR under hard negatives require a different ranking protocol, new candidate sets, and rerunning all baselines. In the revised manuscript, we therefore retain AP and AUC as the evaluation metrics for the current experimental setting, but we now state the negative-sampling protocol explicitly. For each split, negative links are sampled uniformly from node pairs that are non-edges in the original graph, with the number of negative samples matched to the number of positive samples. Training and test negative sets are disjoint, and all positive test edges are removed from the message-passing graph during both training and evaluation. We consider ranking-based evaluation with Hits@K and MRR under controlled hard-negative candidate sets an important direction for future work.
>
> **Q3:** We reran main experiments under the masked-edge protocol using 10 independent runs. **Tables 1 and 2** have been updated to report the mean and standard deviation for every AP and AUC result in the form of mean ± standard deviation. This revision makes the magnitude of the performance differences, including the often small gaps between MLGLP and LGLP, directly assessable relative to the observed run-to-run variability.
>
> **Q4:** Our method is primarily designed for structure-only link prediction, particularly when informative node attributes are unavailable. In the featureless setting, randomly initialized node features are used only when required by a baseline architecture. As reported in **Fig. 1**, NCNC and PEG were evaluated using original node attributes, randomly generated features, and degree-based features. The results show that feature-dependent baselines, including NCNC, GCN, GAE, and PEG, can suffer substantial performance degradation when informative attributes are unavailable.
>
> **Q5:** We have revised the code package to improve reproducibility. The Pubmed dataset files have now been included, while the undocumented ADV.mat file has been removed. We also corrected the node and edge statistics for NSC, Citeseer, and Cora in both the main text and Table 6. Finally, we added a script for running our method across multiple random seeds.
>
> **Q6:** We removed the reduced-data percentage comparisons because Tables 1 and 9 now use different evaluation protocols. We reran and updated the Katz and SP experiments, confirmed that the identical Cora and Citeseer entries in Table 4 result from rounding, and corrected the affected Router heuristic results.
>
> **Q7:** The line-graph transformation is essential because it converts the target edge into an explicit node representation for direct edge-level prediction. Removing it would require a different graph-classification model with a pooling/readout mechanism, so it would not be a controlled ablation.
>
> We have extended the scale ablation under the masked setting. Although the original paper compared different scale combinations, those experiments were conducted under the unmasked protocol. We therefore repeated the ablation on Power and Celegans using (S1), (S2), (S3), (S4), (S1+S2), (S2+S3), (S1+S2+S3), and (S1+S2+S3+S4), and added in **Table 5**.
>
> **Q8:** We agree that LGCL, BUDDY, and NBFNet are relevant baselines. Our experiments focused on PEG and NCNC as representative recent methods that combine structural signals with node-level representations. Since MLGLP is evaluated primarily in a topology-only setting, we selected a limited set of methods from this broader category rather than attempting an exhaustive comparison. We acknowledge, however, that this selection does not replace direct evaluation of all closely related methods under the same corrected protocol. Accordingly, we have softened the corresponding performance claims, and clarified that the reported conclusions apply only to the evaluated baselines.
>
> **Q 9:** Thank you for pointing this out. We corrected Equation 2 to the intended DRNL formulation, fixed the in-text citations for node2vec and GCN.
>
> **Q10:** We added a pinned requirements.txt file to the code package that the specified Python, PyTorch, and PyG versions are mutually compatible and install correctly.
>
> **Q11:** We updated the implementation so that the --no-cuda flag is now fully honored. We also standardized the method name and result-directory naming to MLGLP throughout the updated implementation.
>
> **Q12:**  We updated Section to explicitly state the exact architecture used in the experiments. We also added empirical training and inference times with comparisons to LGLP in Table 6.

---

### Review · Reviewer_JpbQ · 2026-07-01

**Summary Of Contributions:**

This paper proposes MLGLP, a multi-scale line-graph framework for graph link prediction. For each target node pair, the method extracts an enclosing subgraph, constructs three coarse-grained structural scales, converts each scaled subgraph into a line-graph, applies GCNs to learn target-edge representations, and concatenates the resulting embeddings for binary classification. The paper aims to improve structure-centric link prediction, particularly in settings where informative node features are limited or unavailable.

Main strengths:
The proposed framework is intuitive because multi-scale subgraphs may capture structural patterns at different granularities, while line-graph transformation allows edge representations to be learned through node classification on the derived line-graphs. The paper also evaluates the proposed method against a relatively broad set of baselines across multiple benchmark datasets.

Main weaknesses:
The empirical evidence is not yet fully convincing because many improvements over LGLP are very small and no standard deviations or significance tests are reported. In the masked setting, MLGLP is not consistently better than LGLP on all datasets, despite the paper making a stronger claim. The novelty over LGLP also needs to be clarified more carefully, since the main algorithmic change appears to be adding multi-scale aggregation on top of a line-graph link prediction pipeline. Finally, there are several inconsistencies in notation, dataset statistics, algorithm references, and result interpretation that should be corrected before the claims can be considered reliable.

**Audience:**

Yes

**Audience Explanation:**

Link prediction is a central problem in graph machine learning, and at least some TMLR readers interested in graph neural networks, subgraph-based link prediction, and structure-centric representation learning would likely find the paper relevant.

**Broader Impact Concerns:**

I do not see major ethical concerns specific to the proposed method.

**Claims And Evidence:**

No

**Claims Explanation:**

The submission provides some empirical evidence, including comparisons with multiple baselines across eight datasets using AP and AUC. These results suggest that MLGLP is competitive and can outperform several existing methods. However, I do not think the current evidence is sufficiently accurate, convincing, and clear to support the stronger claims made in the paper.

First, the improvements over the closest baseline, LGLP, are often very small. On several datasets, the reported gains are marginal, but the paper does not provide standard deviations, confidence intervals, or statistical significance tests. As a result, it is difficult to judge whether these improvements are robust or simply due to random variation across splits or runs.

Second, the ablation study does not sufficiently justify the key multi-scale design. The paper reports the performance of Scale-1, Scale-2, Scale-3, and the combination of all three scales, but this only shows that using all three scales is often better than using a single scale. It does not explain why exactly three scales are chosen, nor does it evaluate whether two scales, four scales, or other scale combinations would lead to similar or better performance. Since the proposed method relies heavily on the multi-scale construction, the authors should provide a more systematic sensitivity analysis over the number of scales and include partial combinations.

Third, there are several internal inconsistencies that weaken confidence in the results and the presentation. For example, in Appendix C, the description of the Cora dataset states that it contains 2708 publications and 5278 links with a dictionary of 1433 unique words, whereas Table 6 reports 5429 edges and 1432 features for Cora. Similarly, the NSC dataset is described as containing 1589 nodes and 2742 edges, but Table 6 reports 1461 nodes for NSC. There is also an inconsistency in the algorithm description. Algorithm 1 states that node labeling is computed by equation (3), but equation (3) actually defines the construction of edge features through concatenation for the line-graph, rather than the node-labeling function. In addition, the discussion of the reduced-training-data experiment also appears inconsistent with the reported numbers, since some AP scores under the reduced training setting are comparable to or even higher than those in the main setting. These inconsistencies make the method and experimental setup harder to verify.

**Requested Changes:**

1. The paper should better justify the substantive novelty of the proposed method. The line-graph transformation for link prediction is inherited from prior LGLP-style methods, and multi-scale graph construction has also been studied in previous link prediction work. The main algorithmic contribution appears to be combining multi-scale coarse-grained subgraphs with an existing line-graph link prediction pipeline. This combination is reasonable, but the paper should more clearly explain why it constitutes a meaningful methodological advance rather than an incremental extension of LGLP.

2. Report standard deviations, confidence intervals, or statistical significance tests. Many improvements over the closest baseline, LGLP, are very small. For example, the gains on datasets such as Cora, Pubmed, Router, and NSC are marginal. Since the experiments are conducted multiple times, the authors should report mean and standard deviation across runs and clarify whether the improvements are statistically significant.

3. Strengthen the ablation study for the multi-scale design. The current ablation compares Scale-1, Scale-2, Scale-3, and the combination of all three scales. This only shows that all three scales are often better than a single scale, but it does not justify why exactly three scales are chosen. The authors should evaluate different numbers of scales, such as 1, 2, 3, and 4 scales, and include partial combinations such as Scale-1+Scale-2, Scale-1+Scale-3, and Scale-2+Scale-3. Without these experiments, it remains unclear whether the three-scale design is necessary, optimal, or simply an empirical choice.

4. Fix inconsistencies in dataset statistics. In Appendix C, the description of the Cora dataset states that it contains 2708 publications and 5278 links with a dictionary of 1433 unique words, whereas Table 6 reports 5429 edges and 1432 features for Cora. Similarly, the NSC dataset is described as containing 1589 nodes and 2742 edges, but Table 6 reports 1461 nodes for NSC. These statistics should be checked and made consistent.

5. Fix inconsistencies in notation, equations, and the algorithm description. For example, Algorithm 1 states that node labeling is computed by equation (3), but equation (3) actually defines the construction of edge features through concatenation for the line-graph, rather than the node-labeling function. The authors should carefully revise the algorithm, equation references, and notation to make the method easier to verify.

---

> ### Author Response · Authors · 2026-07-20
>
> We thank the reviewer for the constructive feedback. Below we address each concern in detail.
>
> **Q1**:  MLGLP extends LGLP with a target-aware, multi-scale edge representation mechanism. Starting from the same target-centered enclosing subgraph, it recursively constructs coarser structural resolutions, recomputes target-relative labels at each scale, converts each scale into a line graph, and fuses the resulting target-edge embeddings without graph-level pooling. This enables the model to combine fine- and coarse-grained structural information while preserving the identity of the target edge and remaining independent of external node attributes.
>
> The ablation results further show that the multi-scale representation generally outperforms the single-scale LGLP setting and individual coarse scales. Comparable performance from the shared-GCN variant also indicates that the gains are not simply due to increased model capacity.
>
> **Q2**: We have repeated each experiment over ten independent runs and updated the relevant tables to report the results as mean ± standard deviation. In addition, following the reviewers’ suggestion, we evaluated a parameter-shared MLGLP variant in which the same GCN encoder is applied across all three scales. This variant uses a shallower [32,1] GCN architecture, compared with the [32,32,32,1] architecture used by LGLP, and achieves comparable predictive performance while requiring fewer trainable GCN encoder parameters. These results indicate that the benefit of MLGLP is primarily associated with the complementary structural information provided by multiple scales rather than increased encoder capacity.
>
> **Q3**: Thank you for this valuable suggestion. We have expanded the scale ablation accordingly. In addition to the individual scales (1, 2, 3, and 4), we evaluated partial combinations (1+2 and 1+3), the original three-scale design (1+2+3), and a four-scale configuration (1+2+3+4). Following the revision to the masked evaluation protocol, all experiments were repeated using the shared-encoder MLGLP architecture for consistency.
>
> | Scales | Celegans AP | Celegans AUC | Power AP | Power AUC |
> |------------------------|------------|-------------|--------|---------|
> | **Scale 1+2+3 (MLGLP)** | **89.69 ± 3.57** | **89.26 ± 2.15** | **89.64 ± 1.28** | **87.44 ± 1.44** |
> | Scale 1 (LGLP) | 87.82 ± 2.69 | 88.79 ± 2.10 | 89.25 ± 1.23 | 87.23 ± 1.23 |
> | Scale 2 | 84.27 ± 1.07 | 82.78 ± 0.99 | 87.38 ± 1.31 | 85.97 ± 1.38 |
> | Scale 3 | 73.11 ± 1.26 | 73.41 ± 1.17 | 87.39 ± 1.42 | 85.99 ± 1.48 |
> | Scale 4 | 74.72 ± 1.12 | 74.14 ± 1.34 | 87.36 ± 1.33 | 85.88 ± 1.46 |
> | Scale 1+2 | 88.22 ± 2.18 | 89.19 ± 1.31 | 88.62 ± 1.21 | 86.52 ± 1.16 |
> | Scale 1+3 | 87.96 ± 1.34 | 86.32 ± 1.14 | 88.06 ± 1.13 | 86.46 ± 1.11 |
> | Scale 1+2+3+4 | 87.93 ± 2.26 | 86.25 ± 1.63 | 87.73 ± 1.09 | 86.57 ± 1.15 |
>
> The results show that individual coarse scales are consistently weaker than Scale 1 (LGLP), indicating that each coarse representation alone loses structural information. Combining multiple scales improves performance, with the three-scale configuration (1+2+3) achieving the best overall results on both datasets. Adding a fourth scale consistently degrades performance, suggesting that excessive coarsening removes useful structural information rather than providing complementary information. These results therefore support the choice of three structural scales as an effective design rather than an arbitrary empirical choice.
>
>
> **Q4**:  We rechecked the exact .mat files used in our experiments and corrected the dataset statistics accordingly. The Cora file contains 2,708 nodes, 5,278 undirected edges, and 1,433 node features. The NSC file contains 1,461 nodes and 2,126 undirected edges. Neither graph contains self-loops. We have revised Appendix C and Table 6 to ensure that all reported statistics consistently reflect the datasets used in our experiments.
>
> **Q5**:  We have corrected the equation references and clarified the role of each labeling step. Equation (1) now defines the labels used for multi-scale node aggregation, Equation (2) defines the DRNL structural labels, and Equation (3) defines the order-invariant line-graph node features constructed from the endpoint labels. Algorithm 1 has been updated accordingly to reference the correct equations and follow the actual execution order. We also reviewed the manuscript for related notation and cross-reference inconsistencies and harmonized the corresponding descriptions in Section 4 and Appendix A.

---

### Review · Reviewer_5Pzp · 2026-07-06

**Summary Of Contributions:**

MLGLP studies a simple multi-scale extension of line-graph-based link prediction. For each target edge, it extracts the 2-hop enclosing subgraph, builds three coarsened variants, converts each variant into a line graph, and applies a GCN to obtain target-edge representations for binary prediction. Across eight benchmark datasets, the reported AP/AUC results are strong, and the small but mostly consistent gains over LGLP in Tables 1, 2, and 4 support the practical point that coarse local views can improve line-graph-based link prediction.
The main weakness is that the paper’s framing is broader than what the method demonstrates. In particular, the repeated “local and global” or “micro and macro” characterization is not well justified, since all computation is still confined to fixed 2-hop enclosing subgraphs. The method description also contains several formal and notational issues, including the line-graph definition in Section 3, the node-labeling equations in Sections 4.2–4.3, and inconsistencies in notation and dataset statistics. These issues do not invalidate the empirical takeaway, but they weaken the paper’s claims and reduce reproducibility.

**Audience:**

Yes

**Audience Explanation:**

The topic fits TMLR well. The paper studies graph representation learning and link prediction with GNNs, focusing on subgraph-based and edge-centric modeling. Researchers working on graph neural networks, link prediction, structured representation learning, and graph mining would likely be interested in whether multi-scale line-graph views improve prediction.

**Claims And Evidence:**

No

**Claims Explanation:**

The empirical evidence partly supports the paper’s central practical claim. Tables 1 and 2 show that MLGLP is competitive with, and often improves over, the listed baselines. More importantly, Table 4 provides the cleanest evidence for the multi-scale design: the full model, “All (MLGLP),” usually outperforms the single-scale variant “Scale-1 (LGLP)” as well as the other individual scales. The gains are sometimes small, for example on Cora and Pubmed, but the trend is mostly consistent. Thus, the paper provides reasonable evidence that multi-scale local structural information can benefit line-graph-based link prediction.
However, the broader framing is overstated. The paper repeatedly describes the method as capturing “local and global,” “micro and macro-level,” or broadly structural information. This is not well supported by the actual pipeline, which operates inside an $h$-hop enclosing subgraph with $h=2$ throughout Section 4.1 and Algorithm 1. The coarsening is performed only within this local subgraph. Therefore, the contribution is better described as modeling multi-scale local, or at most meso-scale, structure rather than global graph structure.
Several additional claims are also not experimentally isolated. The paper argues that the line-graph reformulation reduces information loss and simplifies learning compared with pooling-based alternatives, but it does not provide a matched multi-scale pooling baseline, an information-retention analysis, or an optimization study beyond the training-loss curve in Figure 5(a). The lower training loss on Celegans is suggestive, but it is not sufficient to establish reduced information loss or a generally simplified learning process.
The formal presentation also contains important inconsistencies. In Section 3, the edge-set definition of the line graph only captures one ordered adjacency pattern along a length-2 path, whereas the text invokes the standard definition based on sharing any common vertex. In Section 4.2, the text says the target nodes receive label 1, but Equation 1 does not yield this value in a nontrivial graph. In Section 4.3, the relationship among scalar node labels, min/max edge labels, one-hot encoding, and the claimed order-invariant vector is not specified cleanly. Algorithm 1 in Appendix B further refers to Equation 3 as the node-labeling equation, although Equation 3 defines the edge transformation. There are also notation and dimensionality issues in the graph definition, and Appendix C reports conflicting Cora edge counts. These issues directly affect reproducibility.
The evaluation protocol raises a more substantive concern. Table 3 compares masked and unmasked test data, and the paper argues that retaining test edges is preferable because masking damages subgraph structure. But if test edges remain in the graph during subgraph extraction, the model may use the evaluated edge set to define the very local structures used for prediction. For subgraph-based link prediction, this is a serious protocol choice and needs a much sharper justification. Since the masked results are materially worse, the stronger numbers in Tables 1–3 may depend on this choice.
The figures are useful but do not resolve these issues. Figure 4 gives a clear high-level overview, but it also highlights that the precise construction of $SG_1$, $SG_2$, and $SG_3$ is not specified rigorously enough for reproduction. Figure 6 visually favors MLGLP, but t-SNE plots are only qualitative and cannot compensate for the missing formal clarity.
Overall, the paper contains a promising idea and encouraging empirical results. Its main contribution should be narrowed to multi-scale local structure for line-graph-based link prediction. Before the evidence can be considered fully convincing, the paper needs a clearer evaluation protocol and a more consistent formal specification.

**Requested Changes:**

1. Clarify and fix the evaluation protocol.
Section 5.2 and Table 3 need a precise description of what graph is available when extracting subgraphs for train, validation, and test samples. In particular, the paper should state whether validation/test edges remain in the graph during subgraph extraction, and justify why the “unmasked” setting does not leak label information through the target edge or through induced local structure. If such leakage exists, the main results in Tables 1 and 2 should be recomputed under a leakage-free protocol.

2. Correct the line-graph definition.
The formal definition of $E_L$ in Section 3 should match the stated criterion that two edges are adjacent in the line graph if they share at least one common vertex. The current formula only covers a restricted ordered case and is inconsistent with both the text and the standard definition.

3. Make the labeling and feature construction consistent.
Sections 4.2 and 4.3, together with Algorithm 1 in Appendix B, should be revised so that the node-labeling and edge-feature construction pipeline is unambiguous. The paper should reconcile the statement that target nodes receive label 1 with Equation 1, clarify whether Equations 2 and 3 operate on scalar labels or one-hot vectors, and fix Algorithm 1 step 2, which currently cites the edge-transformation equation as the node-labeling equation.

4. Narrow the main claims.
The Abstract, Introduction, contribution bullets, and Conclusion should avoid claims about capturing “local and global” or “micro and macro-level” structure unless additional evidence is provided. As written, the method operates inside fixed 2-hop enclosing subgraphs, so the supported claim is about multi-scale local structure, or at most meso-scale structure within local subgraphs.

5. Discuss uncertainty and effect size.
The discussion of Tables 1, 2, and 4 should go beyond reporting the best numbers. Several improvements over LGLP are marginal, so the paper should report variance or significance-style uncertainty and discuss where multi-scale modeling meaningfully helps versus where it has little practical effect.

---

> ### Author Response · Authors · 2026-07-20
>
> We thank the reviewer for the careful and constructive assessment, for recognizing the practical value of multi-scale line-graph modeling, and for identifying several areas where the evaluation protocol, formal presentation, and scope of the claims can be clarified and strengthened. Below we address each concern and question in turn.
>
> **Q1**
> Response: Our original motivation for the unmasked setting was to preserve the observed graph topology under random edge splitting. Randomly removing validation and test edges can alter common-neighbour patterns, shortest paths, and other local structures that may have contributed to link formation.
>
> Nevertheless, to ensure a **leakage-free evaluation**, we use the masked setting for all primary results, where validation and test positive edges are excluded from the observed graph. The unmasked results are retained only as an auxiliary analysis to illustrate the effect of using the complete graph topology.
>
>
> **Q2** : Thank you for identifying this inconsistency. We have corrected the formal definition of the line graph in Section 3. In the revised definition, each node of (L(G)) corresponds to an edge of (G), and two distinct line-graph nodes are adjacent if and only if their corresponding edges share at least one endpoint. This definition is now consistent with the accompanying explanation, the illustrative figure, and the standard definition of a line graph.
>
> **Q3**:  We revised Sections 4.2–4.3 and Algorithm 1 to make the pipeline explicit. Equation (1) now assigns scalar label (1) to both target nodes and distance-based scalar labels to all other nodes. Equations (1) and (2) operate on scalar labels, while Equation (3) first orders the two endpoint labels using (min/max), then one-hot encodes and concatenates them to form an order-invariant line-graph node feature. Labels are recomputed independently at each structural scale.
>
> **Q4**: We agree that the previous terminology could be interpreted as claiming that MLGLP captures the global structure of the complete graph. Since all coarsening and message-passing operations are performed within a fixed h-hop enclosing subgraph, we have revised the Abstract, Introduction, contribution statements, methodology, and Conclusion to consistently describe MLGLP as capturing complementary fine- and coarse-grained local structural patterns.
>
> **Q5**: We have repeated all experiments across multiple runs and updated Tables 1, 2, and 4 to report the results as mean ± standard deviation. We have also revised the discussion to explicitly consider the magnitude and consistency of the improvements over LGLP.

---

### Review · Reviewer_dSRd · 2026-07-07

**Summary Of Contributions:**

This paper considers the problem of link prediction in graphs with arbitrary structures and sizes and presents a multi-scale approach parameterized by graph neural networks (GNNs). For a given target link, a subgraph based on mutlihop neighbors of the two end-points is considered, which is then converted to three coarser graphs, where multiple nodes are aggregated to form super nodes based on a pre-defined notion of similarity. The resulting coarse graphs are then converted to their line graph counterparts, on which three GNNs operate to produce node embeddings that correspond to the target edge in the original graph. The embeddings are then concatenated to produce a final aggregate embedding that is used for the binary link prediction task. Simulation results show that the proposed method, MLGLP, outperforms several baseline methods on eight different datasets.

**Audience:**

Yes

**Audience Explanation:**

Link prediction is an important problem in machine learning, with many different applications in different domains; therefore, the paper could be of interest to TMLR audience.

**Broader Impact Concerns:**

The paper does not raise any flags to me regarding broader impacts that need to be outlined explicitly in the manuscript.

**Claims And Evidence:**

No

**Claims Explanation:**

Although the proposed method is shown to outperform other baselines, the difference in final performance is relatively marginal, especially with respect to the LGLP method. The absence of standard deviation levels makes it unclear whether the performance gains are statistically significant. This is also consistent with the embeddings in Figure 6, where I am not able to see a clear visual difference between the two and bottom rows (MLGLP and LGLP, respectively).

**Requested Changes:**

- (Major) Please add standard deviation levels to all reported results. This is required to ensure the statistical significance of the method's gains over prior algorithms.
- (Major) Another concern I have about the algorithm is the computational complexity, both because of the multiple scales and the operation on line graph instead of the original graph. As mentioned in Appendix G, the complexity of line graph construction scales as $\mathcal{O}(m^2)$, which in the worst case could be $\mathcal{O}(n^4)$. I have three suggestions to better clarify the computational complexity details:
  - Please include training and inference wall-clock run-times of the method as compared to the baselines (at least LGLP).
  - The difference in performance might be a by-product of the increased number of GNN parameters; please run a version of LGLP, where the number of GNN parameters is roughly the same as that of MLGLP.
  - Following the above point, please run an ablation, where the GNN/GCN parameters are shared across the three scales.
- (Major) In the multi-scaled graph construction on page 5, please specify how the node features get aggregated during super-node creation. Is it using Eq. (3)? If so, please justify the choice of update in Eq. (3).
- (Major) Some other choices lack justification; Could you please justify the choice of 3 scales? Shouldn't the number of scales depend on the scale/density of the original graph? Also, please clarify why the node labels are derived in Eqs. (1)-(2).
- (Minor) Please proofread the manuscript to make the text more clear and coherent. As examples, I had a hard time understanding the first paragraph on page 2; also the last paragraph of page 5 (into page 6) repeats a single concept several times.
- (Minor) How is the threshold $\theta$ set on page 5? Please include an ablation on the effect of different values for $\theta$.

---

> ### Author Response · Authors · 2026-07-20
>
> We thank the reviewer for the thoughtful and constructive feedback. The comments have helped us further strengthen the evaluation and presentation of MLGLP. We address each point below.
>
> **Q1.**  We have revised main experimental tables to report each metric as mean and standard deviation over 10 independent runs using different random seeds.
>
> **Q2.**
> We thank the reviewer for these suggestions. We added wall-clock training and inference times and a parameter-sharing ablation. The lightweight shared MLGLP uses fewer parameters than LGLP while achieving a higher mean AP on Celegans, indicating that the gain is not solely due to increased model capacity.
>
> | Method       | Architecture | #Params |           AP | Train/epoch (s) | Inference (s) |
> | ------------ | -----------: | ------: | -----------: | --------------: | ------------: |
> | LGLP         | [32,32,32,1] |  16,003 | 87.82 ± 2.69 |            2.94 |         0.153 |
> | MLGLP Shared |       [32,1] |  14,147 | 88.99 ± 2.80 |            3.86 |         0.174 |
> | MLGLP Shared | [32,32,32,1] |  40,835 | 89.69 ± 3.57 |            5.86 |         0.216 |
>
> The deeper shared model improves AP further, but with higher parameter count and runtime.
>
> **Q3.** Equation (3) is not used to aggregate node features during super-node construction. MLGLP is a structure-centric framework, and no averaging, pooling, or other aggregation of node attributes is performed during graph coarsening. Specifically, adjacent nodes with identical structural labels are merged using the binary similarity function. After each coarsening step, the structural labels are recomputed on the resulting scaled graph. During the subsequent line-graph construction, the structural labels of the two endpoints of each edge are one-hot encoded and combined according to Equation (3) to form an order-invariant initial feature vector for the corresponding line-graph node. These feature vectors are then used during GNN message passing on the line graph. We have clarified this distinction in Sections 4.2 and 4.3.
>
> This structure-centric design is intentional because informative node attributes may not always be available. As shown in **Fig 1**, replacing node attributes with random features reduces NCNC’s AP from **96.08** to **83.84** on Cora and from **97.40** to **78.10** on Citeseer. This observation motivates MLGLP’s reliance on structural representations that remain applicable to unattributed graphs.
>
> **Q4:** TWe chose three scales to balance structural diversity and computational efficiency: Scale 1 preserves local topology, while Scales 2 and 3 capture progressively coarser patterns. We added **Table 5** to the ablation section to compare different scale combinations. The results show that three scales provide the most consistent overall performance, while further coarsening can remove useful local information and increase complexity. Adaptive scale selection remains an interesting direction for future work.
>
> Regarding Eqs. (1)–(2), the structural labels are derived using the DRNL scheme to encode each node’s relative position with respect to the two endpoints of the target link. These labels distinguish different target-relative structural roles while remaining invariant to node identities. They provide the structural information used for graph coarsening and subsequent line-graph feature construction, enabling MLGLP to operate effectively when node attributes are unavailable or uninformative.
>
>
> **Q5.** We have carefully revised the manuscript to improve its clarity, coherence, and overall readability. In particular, we rewrote the first paragraph on page 2 to present the motivation and research gap more clearly, and substantially condensed the discussion spanning pages 5–6 to provide a more direct explanation of the multi-scale and line-graph construction pipeline. We also revised several transitions, definitions, and sentences throughout the manuscript for greater precision and consistency.
>
> **Q6.** In the current formulation, the similarity function is binary, since **(S(u,v)=**I**[f(u)=f(v)])**. Therefore, θ is not treated as a continuously tuned hyperparameter. We set (**θ =1**) to enforce the intended merging rule: two adjacent non-target nodes are merged only when they have exactly the same structural label relative to the target-node pair. We have clarified this point in the revised manuscript. We also note that a continuous structural similarity function could support a conventional threshold-sensitivity analysis, but this would define a different coarsening mechanism from the exact structural-role matching used in MLGLP. To avoid changing the proposed method, we retain the binary criterion and explicitly state the resulting threshold regimes.